# Actin-related protein 5 functions as a novel modulator of MyoD and MyoG in skeletal muscle and in rhabdomyosarcoma

**Tsuyoshi Morita[1]\*, Ken'ichiro Hayashi[2]**

[1]Department of Biology, Wakayama Medical University, Wakayama, Japan;
[2]Department of RNA Biology and Neuroscience, Osaka University Graduate School of Medicine, Osaka, Japan

**Abstract** Myogenic regulatory factors (MRFs) are pivotal transcription factors in myogenic differentiation. MyoD commits cells to the skeletal muscle lineage by inducing myogenic genes through recruitment of chromatin remodelers to its target loci. This study showed that actin-related protein 5 (Arp5) acts as an inhibitory regulator of MyoD and MyoG by binding to their cysteine-rich (CR) region, which overlaps with the region essential for their epigenetic functions. Arp5 expression was faint in skeletal muscle tissues. Excessive Arp5 in mouse hind limbs caused skeletal muscle fiber atrophy. Further, Arp5 overexpression in myoblasts inhibited myotube formation by diminishing myogenic gene expression, whereas Arp5 depletion augmented myogenic gene expression. Arp5 disturbed MyoD-mediated chromatin remodeling through competition with the three-amino-acid-loop-extension-class homeodomain transcription factors the Pbx1–Meis1 heterodimer for binding to the CR region. This antimyogenic function was independent of the INO80 chromatin remodeling complex, although Arp5 is an important component of that. In rhabdomyosarcoma (RMS) cells, Arp5 expression was significantly higher than in normal myoblasts and skeletal muscle tissue, probably contributing to MyoD and MyoG activity dysregulation. Arp5 depletion in RMS partially restored myogenic properties while inhibiting tumorigenic properties. Thus, Arp5 is a novel modulator of MRFs in skeletal muscle differentiation.

**\*For correspondence:**
tsuyo@wakayama-med.ac.jp

**Competing interest:** The authors declare that no competing interests exist.

## Editor's evaluation

This study describes a novel mechanism by which ARP5 interacts with the master transcription factor MYOD to antagonize its function and thereby the myogenic gene expression program. ARP5 is up-regulated in rhabdomyosarcoma cells and appears to inhibit MYOD function as its depletion partially restores the myogenic program in the tumour cells. ARP5 is therefore a novel regulator of myogenesis in both normal and pathological contexts.

## Introduction

Actin-related protein 5 (Arp5), encoded by *Actr5*, is a nuclear-localized actin-like protein (*Schafer and Schroer, 1999*). Studies have investigated the role of Arp5 in the nucleus as one of the subunits of the ATPase-dependent chromatin remodeling complex INO80 (*Shen et al., 2000*). INO80 regulates various DNA metabolic processes, such as gene expression, DNA replication, and DNA repair by nucleosome sliding (*Poli et al., 2017*). It contains β-actin and three Arp family members (Arp4, Arp5, and Arp8). Arp5 forms the Arp5 module with Ies6 (encoded by *Ino80c*), which is necessary for ATP

hydrolysis and nucleosome sliding by INO80 (*Yao et al., 2015*). However, a few features of Arp5 are unrelated to INO80. In *Arabidopsis*, Arp5 and Ino80, an essential ATPase component of INO80, have common and distinct features in plant growth and development (*Kang et al., 2019*). In addition, Arp5 plays an INO80-independent role in regulating the differentiation of vascular smooth muscle cells (SMCs) (*Morita and Hayashi, 2014*). In rat SMCs, Arp5 expression is strongly inhibited, although INO80 activity is generally necessary for cell growth and proliferation. Arp5 inhibits SMC differentiation by interacting with and inhibiting the SAP family transcription factor myocardin, which is a key regulator for the induction of SMC-specific contractile genes, indicating that low Arp5 expression in SMCs contributes to maintaining their differentiation status. In contrast, in cardiac and skeletal muscles, although Apr5 expression is low, its physiological significance is unclear.

Myogenic regulatory factors (MRFs), such as MYF5, MyoD, MyoG, and MRF4, are skeletal-muscle-specific basic helix–loop–helix (bHLH) transcription factors and master regulators of skeletal muscle development. They recognize a *cis*-regulatory element E-box, usually found in promoter and enhancer regions of muscle-specific genes, as a heterodimer with ubiquitous bHLH proteins of the E2A family (E12 and E47) (*Funk et al., 1991*). MRFs enhance the transcriptional activity of myogenic genes via chromatin remodeling by recruiting the switch/sucrose nonfermentable (SWI/SNF) complex to previously silent target loci (*Roy et al., 2002*; *Ohkawa et al., 2007*). This epigenetic activity depends on a histidine- and cysteine-rich (H/C) region N-terminal to the bHLH domain in MRFs (*Gerber et al., 1997*). This region contains a CL-X-W motif, which is a binding site for the heterodimer of three-amino-acid-loop-extension (TALE)-class homeodomain transcription factors (Pbx1 and Meis/Prep1) (*Knoepfler et al., 1999*). *Funk and Wright, 1992*, reported that E-box elements recognized by MyoG are occasionally flanked by a novel consensus motif TGATTGAC, which was also identified as a binding motif for the Pbx1–Meis/Prep1 heterodimer (*Knoepfler et al., 1999*). Thus, MRFs form a complex with the TALE heterodimer on DNA, leading to the recruitment of chromatin remodelers and an increase in the accessibility of their target loci.

This study reported a novel role of Arp5 in myogenic differentiation of skeletal muscle cells. We identified Arp5 as an inhibitory binding protein for MyoD and MyoG in skeletal muscle and rhabdomyosarcoma (RMS) cells. Results showed that Arp5 competes with the Pbx1–Meis1 heterodimer for interaction with the cysteine-rich (CR) region of MyoD and MyoG and consequently inhibits myogenic differentiation.

## Results

### Arp5 prevents skeletal muscle development by inhibiting MRF expression

Arp5 expression was significantly low in heart, aorta, and especially, hind limb muscle tissues (*Figure 1A*). Besides, Arp5 expression in primary mouse myoblasts significantly decreased with myotube differentiation (*Figure 1B*). The public human transcriptome databases (Human Protein Atlas [HPA], Genotype-Tissue Expression [GTEx], and Functional Annotation of the Mouse/Mammalian Genome 5 [FANTOM5]) showed low *ACTR5* expression in skeletal muscle tissues (*Figure 1—figure supplement 1*). In this study, 5 weeks after injection of the Arp5-AAV6 vector, the muscle fiber thickness significantly reduced compared to the control group (*Figure 1C*). In these atrophic muscles, gene expression levels of MRFs (*Myod1*, *Myog*, and *Myf6*) significantly decreased, accompanied by a decrease in other skeletal muscle markers, such as *Myh4*, *Acta1*, and *Tnni1* (*Figure 1D*).

In C2C12 cells, Arp5 overexpression significantly inhibited the fusion ability of myoblasts and the induction of MyoG and myosin heavy chain (MHC) under the conditions for myotube formation (*Figure 1E and F*). In addition, the induction of *Myod1*, *Myog*, *Myf6*, and *Myh3* was also significantly inhibited (*Figure 1G*). Conversely, gene silencing of Arp5 by short interfering RNA (siRNA) (Arp5-si) increased the expression of these myogenic genes even in the growth medium, although myotube formation was not observed (*Figure 1H*). In primary mouse myoblasts, Arp5 overexpression also decreased the expression of MRFs and *Myh2*, whereas Arp5 silencing increased their expression (*Figure 1I and J*).

Long-term exposure to 5-azacytidine converts C3H 10T1/2 mouse embryo fibroblasts into differentiated skeletal muscle cells with the induction of endogenous MRF expression (*Davis et al., 1987*). There was significant induction of endogenous *Myod1* and *Myog* 7 days after 5-azacytidine treatment,

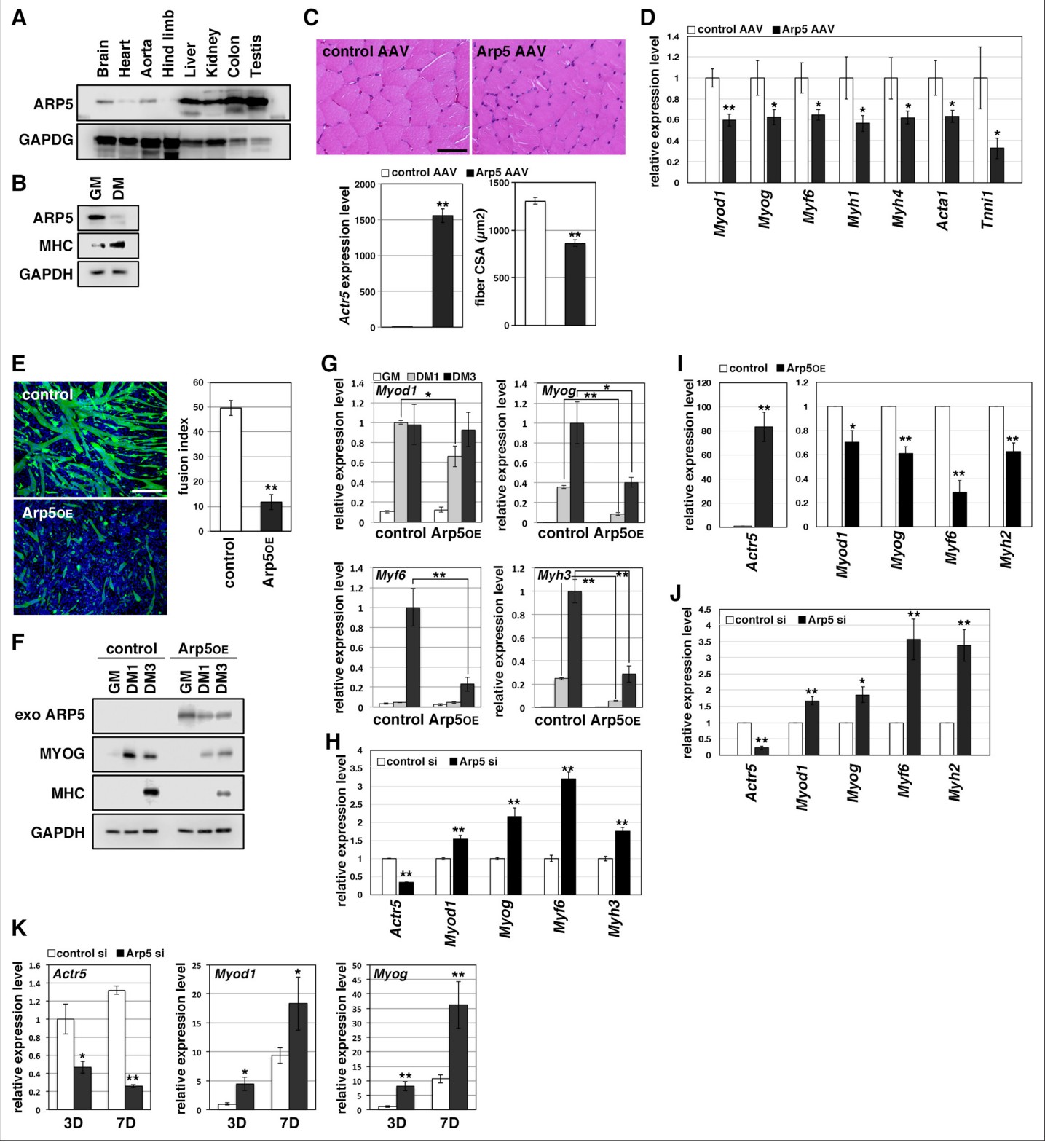

**Figure 1.** Actin-related protein 5 (Arp5) inhibits skeletal muscle differentiation. (**A**) Arp5 expression in mouse tissues. (**B**) Arp5 and myosin heavy chain (MHC) expression in C2C12 cells cultured in growth medium (GM) or differentiation medium (DM). (**C**) Representative images of hematoxylin and eosin (H&E)-stained section of the hind limb muscle from mice injected with control or Arp5-AAV6 vector (upper). Scale bar = 50 µm. Arp5 overexpression was validated by real-time reverse transcription polymerase chain reaction (RT-PCR) (lower left). Muscle fiber cross-sectional area (CSA) were measured in 190 fibers and statistically analyzed (lower right). (**D**) Myogenic gene expression in AAV6-vector-injected hind limb muscles. (**E**) Representative fluorescence

*Figure 1 continued on next page*

*Figure 1 continued*

images of differentiated C2C12 cells transfected with green fluorescent protein (GFP) alone (control) or together with Arp5 (Arp5^OE) (left). Nuclei were visualized by Hoechst 33342. Scale bar = 100 μm. The fusion index was measured on 22 images and statistically analyzed (right). (**F**) Myogenic protein expression in C2C12 cells transfected with control or Arp5 expression vector. The cells were cultured in GM or DM for 1 day (DM1) or 3 days (DM3) after transfection. (**G**) Myogenic gene expression in Arp5-transfected C2C12 cells. (**H**) Myogenic gene expression in C2C12 cells transfected with control or Arp5 short interfering RNA (siRNA). (**I**) Myogenic gene expression in Arp5-transfected mouse primary myoblasts. (**J**) Myogenic gene expression in mouse primary myoblasts transfected with control or Arp5 siRNA. (**K**) Myogenic gene expression in 10T1/2 cells treated with 5-azacytidine. The cells were transfected with control or Arp5 siRNA prior to 5-azacytidine treatment. All statistical data are presented as the mean ± standard error of the mean (SEM). *$p < 0.05$, **$p < 0.01$ (Student's *t*-test).

The online version of this article includes the following source data and figure supplement(s) for figure 1:

**Source data 1.** Raw data used for the statistical analyses presented in *Figure 1*.

**Source data 2.** Original western blot data for *Figure 1A, B and F*.

**Figure supplement 1.** Expression profiles of actin-related protein 5 (*ACTR5*) in human tissues.

and Arp5 silencing significantly augmented *Myod1* and *Myog* induction (*Figure 1K*). These findings show that Arp5 plays an inhibitory role in skeletal muscle development via regulating MRF expression and activity.

## High Arp5 expression contributes to defective myogenic differentiation in RMS

RMS is a common soft-tissue sarcoma developed from skeletal muscle with defective myogenic differentiation. High MyoD and MyoG expression is used in the clinical diagnosis of RMS, but they are not fully active for the induction of their target genes (*Folpe, 2002*; *Keller and Guttridge, 2013*). *ACTR5* expression was significantly higher in human embryonal and alveolar RMS tissues compared with healthy and tumor-adjacent skeletal muscles (*Figure 2A*). Human RMS RD cell line also showed abundant expression of Arp5 at the transcription and translation levels compared to primary human myoblasts (*Figure 2B*). These results suggest that Arp5 overexpression may contribute to MRF dysfunction in RMS.

Microarray analysis on Arp5-silencing RD cells showed that Arp5-si increased the expression of numerous genes involved in muscle function and development (*Figure 2C and D*, *Figure 2—source data 3*). The increased myogenic gene expression was also confirmed by real-time reverse transcription polymerase chain reaction (RT-PCR) (*Figure 2E*). Unlike myoblasts, RD cells have little or no ability to form myotubes, even under serum-free culture conditions, but Arp5 depletion led RD cells to form numerous myotube-like structures with upregulation of myogenic marker proteins (*Figure 2F*).

Arp5 is one of the subunits of INO80, so siRNA (Ies6-si and Ino80-si) also depleted the expression of other subunits (Ies6 and Ino80). The genes altered by the silencing of each of these partly overlapped (*Figure 2—figure supplement 1A, B*); 39.2% and 32.4% of the genes altered by Arp5-si overlapped with those altered by Ies6-si and Ino80-si, respectively. Contrary to Arp5-si, however, Ies6-si and Ino80-si hardly increased the expression of muscle-related genes (*Figure 2C and E*, *Figure 2—figure supplement 1C-E*). These findings show that Arp5 overexpression inhibits the expression of muscle-related genes and, therefore, terminal myogenic differentiation in RMS in an INO80-independent manner.

## Arp5 knockout leads to loss of tumorigenicity of RD cells

In Arp5-knockout RD (Arp5-KO) cells, a few nucleotides downstream of the start codon of *ACTR5* were deleted by CRISPR-Cas9 genome editing, causing Arp5 deletion (*Figure 3A*). Arp5-KO cells showed significant upregulation of many kinds of skeletal muscle-related genes (*Figure 3B*).

Under culture conditions, Arp5-KO cells proliferated as well as parental RD cells (doubling time of wild-type [WT] RD cells [$t_2$(WT)] = 29.7 hr, $t_2$(C39) = 29.8 hr, $t_2$(C45) = 23.8 hr, $t_2$(C67) = 26.7 hr; *Figure 3C*). However, when transplanted subcutaneously into nude mice, they rarely formed tumor nodules differently from parental cells (*Figure 3D*). These rare tumor nodules showed higher expression of muscle-related genes and *CDKN1A*, which encodes a cyclin-dependent kinase inhibitor p21^WAF1/CIP1 controlling cell cycle arrest and myogenic differentiation during muscle development and

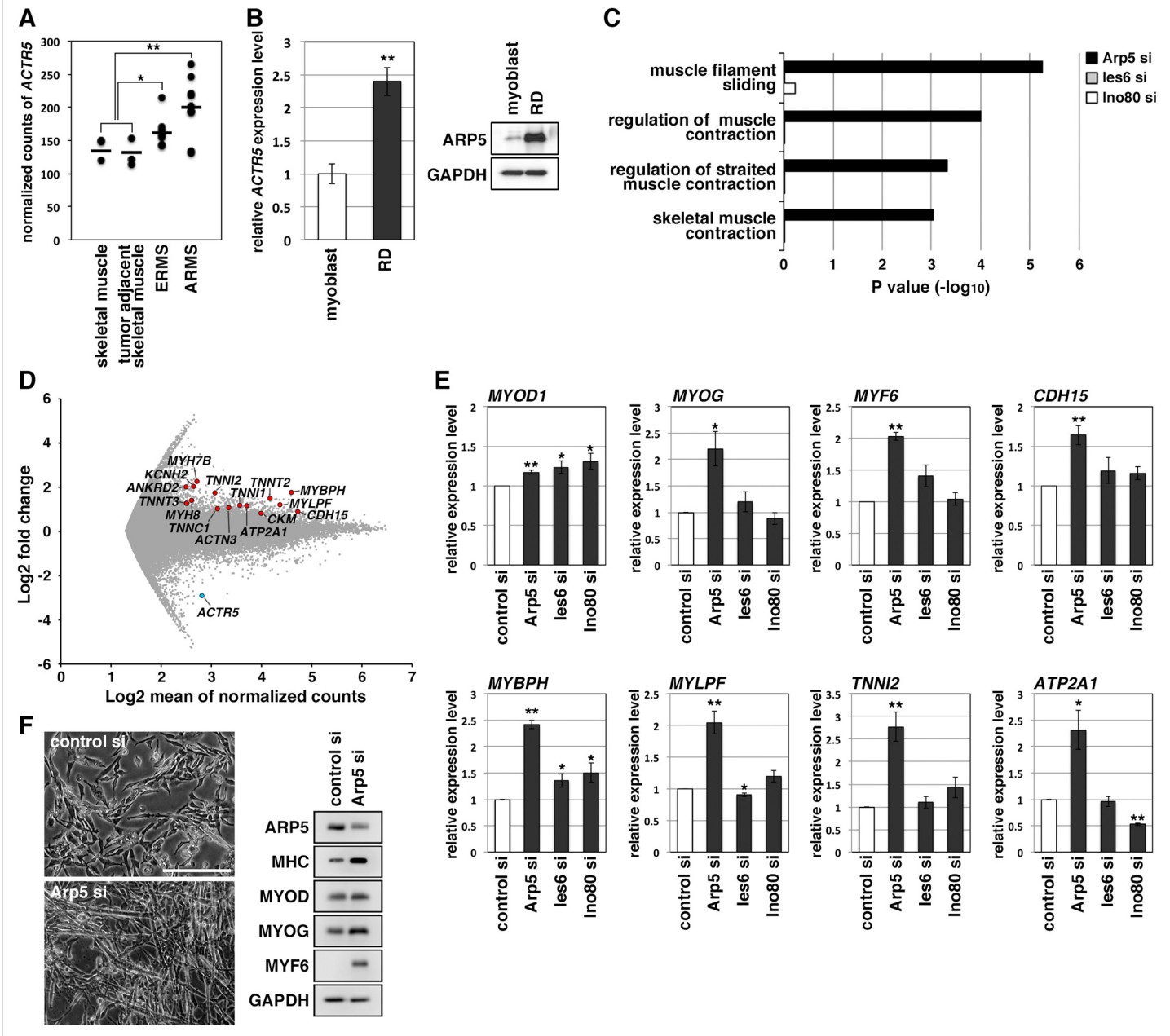

**Figure 2.** Actin-related protein 5 (Arp5) knockdown increases myogenic gene expression in rhabdomyosarcoma (RMS) cells. (**A**) Comparison of *ACTR5* expression in normal skeletal muscle, tumor-adjacent skeletal muscle, embryonal RMS (ERMS), and alveolar RMS (ARMS) from published RNA-Seq data (GSE28511). Bars indicate average expression levels. (**B**) Gene (left) and protein (right) expression of Arp5 in human primary myoblasts and RD cells. (**C**) Enrichment analysis of muscle-related Gene Ontology terms from DNA microarray data on comparison between control-si and Arp5-, Ies6-, or Ino80-si samples. (**D**) MA plot of gene expression level in control-si and Arp5-si RD cells. The positions of Arp5 (blue) and myogenic genes (red) are shown. (**E**) Myogenic gene expression in control-, Arp5-, Ies6-, and Ino80-si RD cells. (**F**) Representative phase-contrast images of control-si and Arp5-si RD cells (left). Scale bar = 100 μm. Myogenic protein expression in these cells (right). All statistical data are presented as the mean ± standard error of the mean (SEM). *p < 0.05, **p < 0.01 (Student's *t*-test).

The online version of this article includes the following source data and figure supplement(s) for figure 2:

**Source data 1.** Raw data used for the statistical analyses presented in *Figure 2*.

**Source data 2.** Original western blot data for *Figure 2B and F*.

**Source data 3.** List of the myogenic genes whose expression level is upregulated by actin-related protein 5 (Arp5) knockdown in RD cells.

**Figure supplement 1.** Expression profiles of genes altered by actin-related protein 5 (*ACTR5*), *IES6*, and *INO80* knockdown.

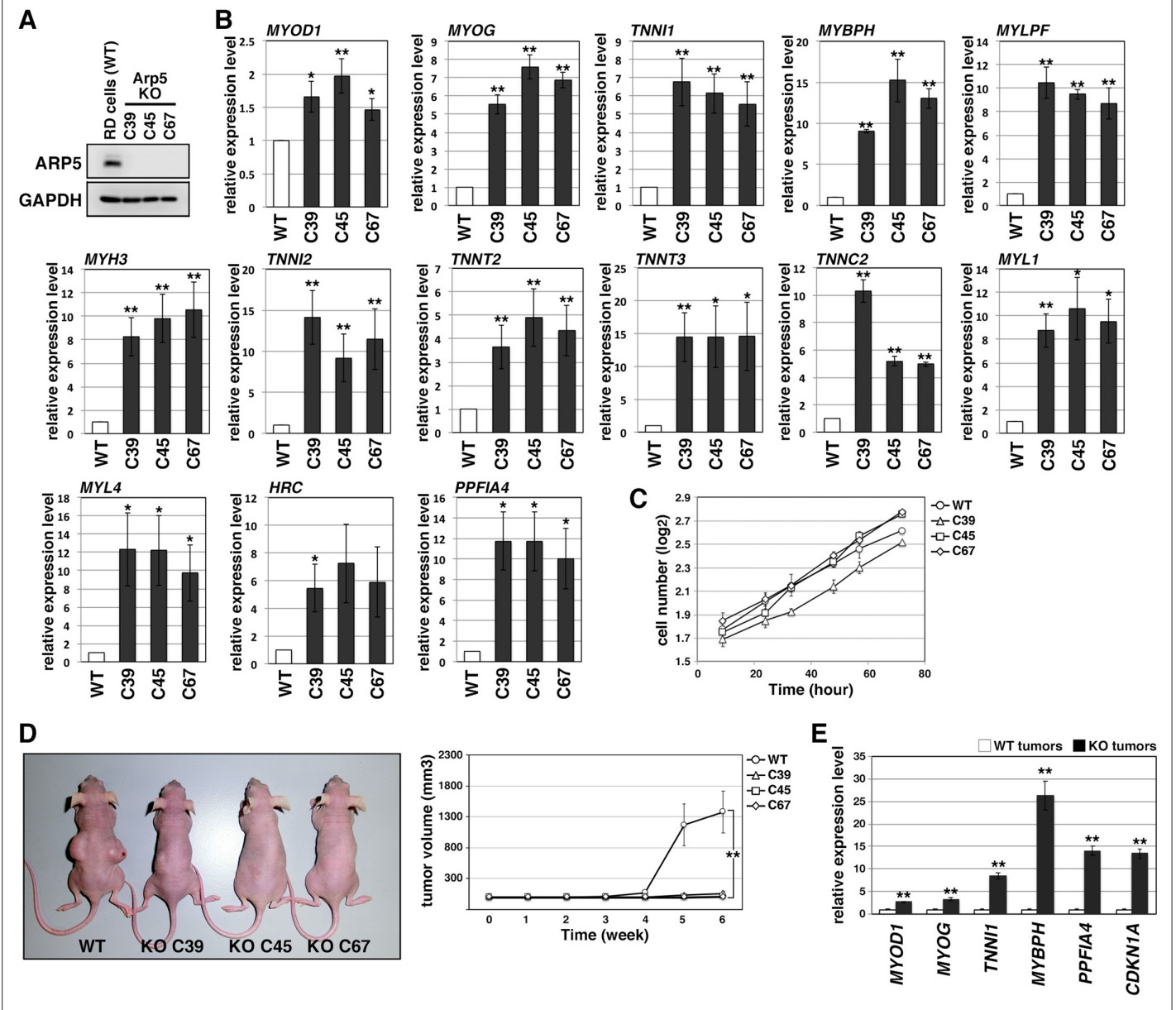

**Figure 3.** Actin-related protein 5-knockout (Arp5-KO) RD clones show increased expression of myogenic genes and decreased tumorigenicity. (**A**) Arp5 expression in three individual clones of Arp5-KO cells (C39, C45, and C67) and their parental RD cells (wild-type [WT]). (**B**) Myogenic gene expression in WT and Arp5-KO cells. (**C**) Growth curve of WT and Arp5-KO cells. (**D**) Xenograft model of WT and Arp5-KO cells in nude mice (left). Tumor volumes measured every week after inoculation and statistically analyzed (right). (**E**) Myogenic gene expression in xenograft tumors. All statistical data are presented as the mean ± standard error of the mean (SEM). *p < 0.05, **p < 0.01 (Student's *t*-test).

The online version of this article includes the following source data for figure 3:

**Source data 1.** Raw data used for the statistical analyses presented in *Figure 3*.

**Source data 2.** Original western blot data for *Figure 3A*.

regeneration (*Halevy et al., 1995*; *Figure 3E*). Thus, Arp5 deletion partially restores the impaired myogenic differentiation potential of RD cells and inhibits their tumorigenesis in vivo.

## Arp5 inhibits MyoD and MyoG activity through binding to their CR region

MRF transcription is regulated by positive autoregulation (*Tapscott, 2005*). To examine whether Arp5 inhibits the activity or transcription of MRF, immunoprecipitation and reporter promoter analysis were performed. Immunoprecipitation showed that Arp5 binds to both MyoD and MyoG but not to the ubiquitous bHLH protein E47 (*Figure 4A*). MyoG was precipitated more efficiently than MyoD with Arp5. We also confirmed the interaction between endogenous Arp5 and MyoD by a co-immunoprecipitaion assay (*Figure 4B*). The amino acid sequences of basic and HLH regions were highly conserved between MyoD and MyoG (*Figure 4—figure supplement 1*), but these domains were not necessary for their interaction (*Figure 4C*). The CR region, a part of the H/C region, was also highly conserved between them (*Figure 4—figure supplement 1*), and CR region deletion mostly abolished the Arp5-binding ability of both MyoD and MyoG (*Figure 4C and D*). MyoG promoter-controlled luciferase reporter assay demonstrated that MyoD strongly enhances MyoG promoter activity, while Arp5 significantly inhibits it (*Figure 4E*). This inhibitory effect of Arp5 was also shown under the condition of Ino80 knockdown (*Figure 4F*). MyoD ΔCR, in which the CR region was deleted, also increased MyoG promoter activity but to a lesser extent, and this activation was completely unaffected by Arp5 (*Figure 4E*), indicating that the direct interaction with MyoD via the CR region is necessary for Arp5 to inhibit MyoD activity.

Ectopic MyoG expression differentiated 10T1/2 cells into MHC-positive myogenic cells with induction of endogenous myogenic marker genes such as *Myod1*, *Myog*, *Ckm*, and *Myh3* (*Figure 4G and H*). Co-expression of Arp5 with MyoG, however, significantly decreased the frequency of MHC-positive cells and significantly inhibited the induction of myogenic genes (*Figure 4G and H*). Thus, Arp5 inhibits MyoD/MyoG activity through direct interaction via their CR region.

## Arp5 competes with Pbx1–Meis1 for binding to the CR region of MyoD/MyoG

The H/C region of MyoD is an essential region for its chromatin remodeling activity and effective induction of target genes (*Gerber et al., 1997*). Pbx1 and Meis1 homeobox proteins are promising candidates as mediators between MyoD and chromatin remodelers because MyoD binds to Pbx1–Meis1 heterodimer via the H/C region (*Knoepfler et al., 1999*). To evaluate the competitive relationship between Arp5 and the Pbx–Meis heterodimer for the MyoD interaction, we measured the relative binding affinities of Arp5, Pbx1, and Meis1 for MyoD. Pbx1 and Arp5 bound to MyoD with moderate affinity—with Kd values of 1.69 ± 0.34 and 4.03 ± 0.72 μM, respectively—whereas the binding affinity of Meis1 for MyoD was rather weak (Kd = 10.48 ± 1.38 μM) (*Figure 5A*). Co-immunoprecipitation assays revealed physical interaction between exogenously expressed MyoD and Pbx1–Meis1 in vivo, and this interaction was actually interrupted by Arp5 (*Figure 5B*).

The *MYOG* locus has been most analyzed for MyoD-mediated chromatin remodeling. The proximal promoter region of *MYOG* reportedly contains two flanking sequences, Pbx1–Meis1 heterodimer-binding motif (PM) and noncanonical E-box (ncE) (*Berkes et al., 2004*; *Figure 5C*). When this regulatory region (PM/ncE-containing region 1 [PME1]) was fused to the TATA minimal promoter, the PME1–TATA construct was activated by the combination of Pbx1, Meis1, and MyoD (*Figure 5D*, left). Upstream of PME1, a novel predicted regulatory region containing PM and E-box motifs (PME2), was identified (*Figure 5C*). A pull-down assay using DNA-probe-conjugated beads of the PMEs demonstrated that the Pbx1–Meis1 heterodimer recognizes the PM motif of both PME1 and PME2 and that this interaction completely disappears by disruption of the PM motif via mutagenesis (*Figure 5E*). The PME2–TATA construct was also synergically activated by MyoD and the Pbx1–Meis1 heterodimer (*Figure 5D*, right). Thus, both PME1 and PME2 are functional *cis*-regulatory elements for MyoD and the Pbx1–Meis1 heterodimer.

The DNA–protein pull-down assay also demonstrated that the MyoD–E47 heterodimer weakly binds to PME1 and PME2 DNA probes, which is significantly augmented by co-incubation with the Pbx1–Meis1 heterodimer (*Figure 5F*). The Pbx1–Meis1 heterodimer was constantly associated with the PMEs regardless of whether MyoD-E47 was present. Arp5 interrupted the augmented interaction

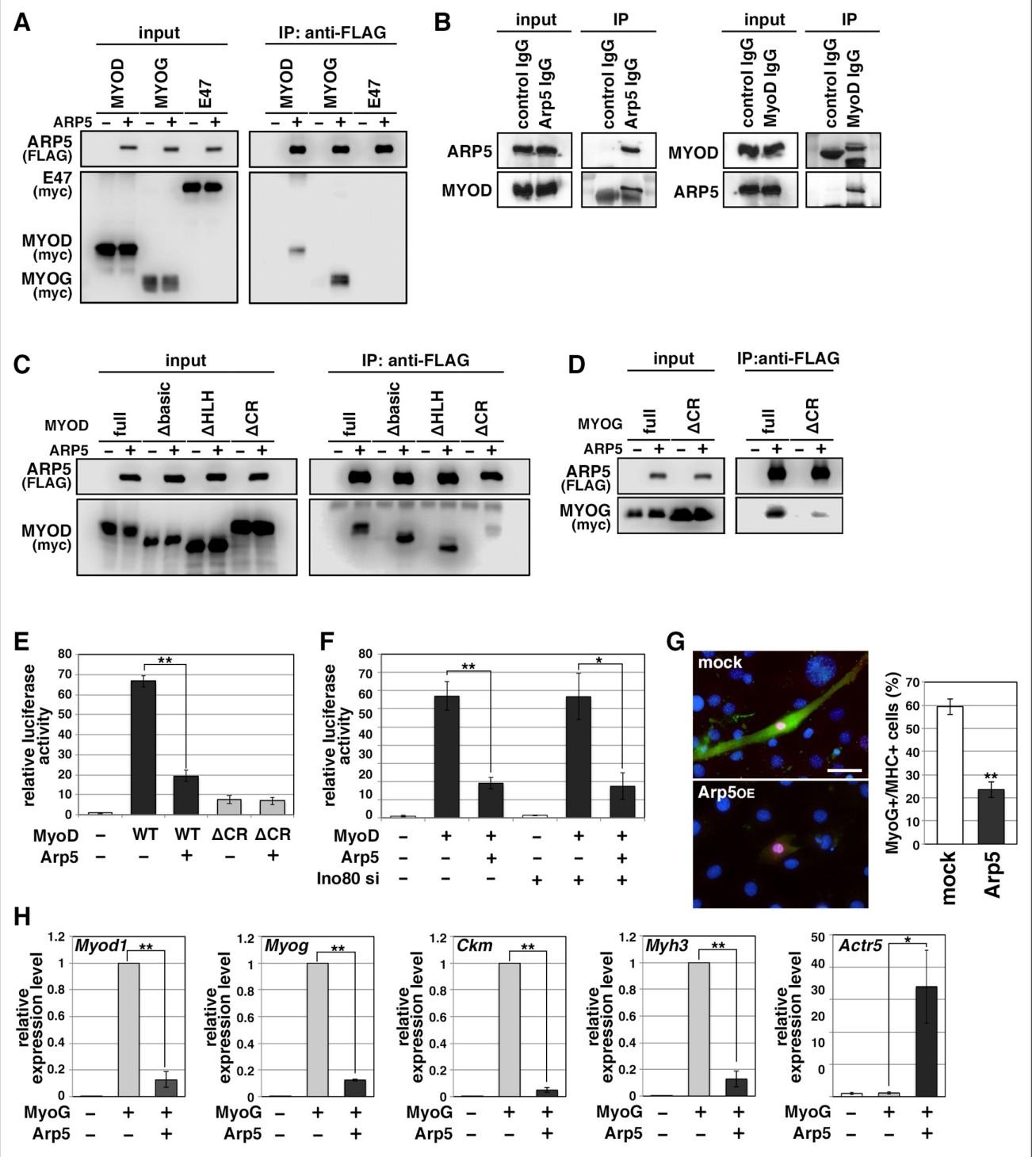

**Figure 4.** Actin-related protein 5 (Arp5) inhibits the activity of MyoD and MyoG through direct interaction. (**A**) Co-immunoprecipitation assay between exogenous Arp5 and basic helix–loop–helix (bHLH) transcription factors MyoD, MyoG, and E47. (**B**) Co-immunoprecipitation assay between endogenous Arp5 and MyoD in RD cells. (**C**) Co-immunoprecipitation assay between exogenous Arp5 and truncated series of MyoD. (**D**) Co-immunoprecipitation assay between exogenous Arp5 and the cysteine-rich (CR)-region-deleted MyoG. (**E**) *Myog* promoter-controlled luciferase reporter assay in C2C12 cells. (**F**) *MYOG* promoter-controlled luciferase reporter assay under the *Ino80*-depleted condition. (**G**) Representative fluorescence images of 10T1/2 cells transfected with MyoG and Arp5 (left). The cells were immunostained with anti-MyoG (red) and anti-myosin heavy chain (MHC, green) antibodies. Nuclei were visualized by Hoechst 33342 (blue). Scale bar = 50 μm. The percentage of MyoG+ MHC+ double-positive cells in MyoG+-positive cells was calculated in 786 cells and statistically analyzed (right). (**H**) Endogenous myogenic gene expression in 10T1/2 cells transfected with MyoG and Arp5. All statistical data are presented as the mean ± standard error of the mean (SEM). *p < 0.05, **p < 0.01 (Student's *t*-test).

*Figure 4 continued on next page*

*Figure 4 continued*

The online version of this article includes the following source data and figure supplement(s) for figure 4:

**Source data 1.** Raw data used for the statistical analyses presented in *Figure 4*.

**Source data 2.** Original western blot data for *Figure 3A, B, C and D*.

**Figure supplement 1.** Sequence alignment between human MyoD and MyoG proteins.

of MyoD–E47 with the PMEs but did not affect the interaction of the Pbx1–Meis1 heterodimer with the PMEs (*Figure 5F*).

We also identified the predicted PME element in the proximal promoter region of *MYF6*, which contains Pbx- and Meis-binding motifs separated by six nucleotides (acTGATgctccaTGACag) close to noncanonical E-boxes (*Figure 6A*). *Jacobs et al., 1999*, reported that this type of gapped PM site is also functional for the Pbx1–Meis1-binding site, and we observed significant interaction between the Pbx1–Meis1 heterodimer and *MYF6*'s PM motif (*Figure 6B*). Similar to PMEs in the *MYOG* promoter, MyoG bound to *MYF6*'s PME synergically with the Pbx1–Meis1 heterodimer, while Arp5 interrupted this binding (*Figure 6C*). These findings show that Arp5 attenuates MyoD/MyoG recruitment to PME-containing myogenic enhancer regions by disturbing the interaction between MyoD/MyoG and the Pbx1–Meis1 heterodimer.

## Arp5 prevents Pbx and Brg1 recruitment to MyoD/MyoG target loci

During MyoD-mediated chromatin remodeling, the Pbx1–Meis1 heterodimer is believed to be involved in the recruitment of Brg1-based SWI/SNF chromatin remodeling complex to MyoD target loci as a pioneering factor (*de la Serna et al., 2001*; *de la Serna et al., 2005*). Therefore, we investigated the change in MyoD target gene expression by Arp5-si in terms of Brg1 dependency. Arp5-si upregulated Brg1-dependent genes to a larger extent compared to Brg1-independent genes (1.77-fold versus 1.38-fold, p = 0.02; ). This Brg1 dependency was seen more clearly by comparing the effect of Arp5-si versus Ies6-si and Ino80-si: Arp5-si more effectively increased Brg1-dependent gene expression compared with Ies6-si and Ino80-si, whereas Brg1-independent gene alteration was comparable between them (*Figure 7A*). These data raise the probability that Arp5—and not Ies6 or Ino80—affects the expression of MyoD target genes in a Brg1-dependent manner.

To determine the role of Arp5 in regulating Brg1 recruitment to MyoD target loci, we performed chromatin immunoprecipitation (ChIP) assay using Arp5-KO RD cells. MyoD, MyoG, Pbx, and Brg1 were recruited to the proximal promoter region of *MYOG*. MyoD, MyoG, and Brg1 significantly accumulated in Arp5-KO cells compared with parental RD cells (*Figure 7B*). Arp5-KO increased Pbx1 recruitment to a lesser extent. ChIP assay against the enhancer regions of Brg1-dependent myogenic genes, *TNNI1*, *MYBPH*, and *PPFIA4*, using public ChIP-Seq and DNase I hypersensitive site (DHS)-Seq databases identified Pbx1- and MyoD-binding regions in their promoter and intronic regions; these regions were DNase I hypersensitive and positive for H3K27Ac, which are markers for the identification of active enhancer regions (*Figure 7—figure supplement 1*). Pbx–Meis-binding motifs actually exist close to canonical and noncanonical E-box motifs in the focused areas (*Figure 7—figure supplement 1*). ChIP data showed that all the proteins of interest were recruited to the predicted enhancer regions and that Arp5-KO significantly augmented MyoD, MyoG, and Brg1 accumulation in these regions (*Figure 7B*). We also investigated the involvement of Arp5 in chromatin structure alteration of the MyoD/MyoG target loci. DNase I accessibility to the above-mentioned regions was compared between Arp5-KO and parental RD cells (*Figure 7C*). The accessibility of all the target regions was significantly higher in Arp5-KO RD cells, suggesting that Arp5 prevents the recruitment of the Brg1 SWI/SNF complex to the target loci and therefore decreases chromatin accessibility.

Finally, we performed a genome-wide ChIP-Seq analysis of MyoD using Arp5-KO and WT RD cells. Meta-gene profiles and density heatmap of the ChIP-Seq reads showed the enriched binding of MyoD near the transcription start sites (TSS) in both cells (*Figure 7D and E*). The heatmap showed somewhat higher concentrations of Arp5-KO reads compared to WT reads near the TSS of several genes. MACS2 analysis called 4387 and 2502 peaks in Arp5-KO and WT RD cells, respectively. These peaks were primarily located in intergenic regions (44.8% in WT peaks and 31.8% in Arp5-KO peaks) and introns (39.2% and 46.0%) (*Figure 7F*). The percentage of peaks located near the TSS, including promoter regions, 5'UTR, exon1, and intron1, was slightly increased in Arp5-KO cells (34.7%) compared to WT

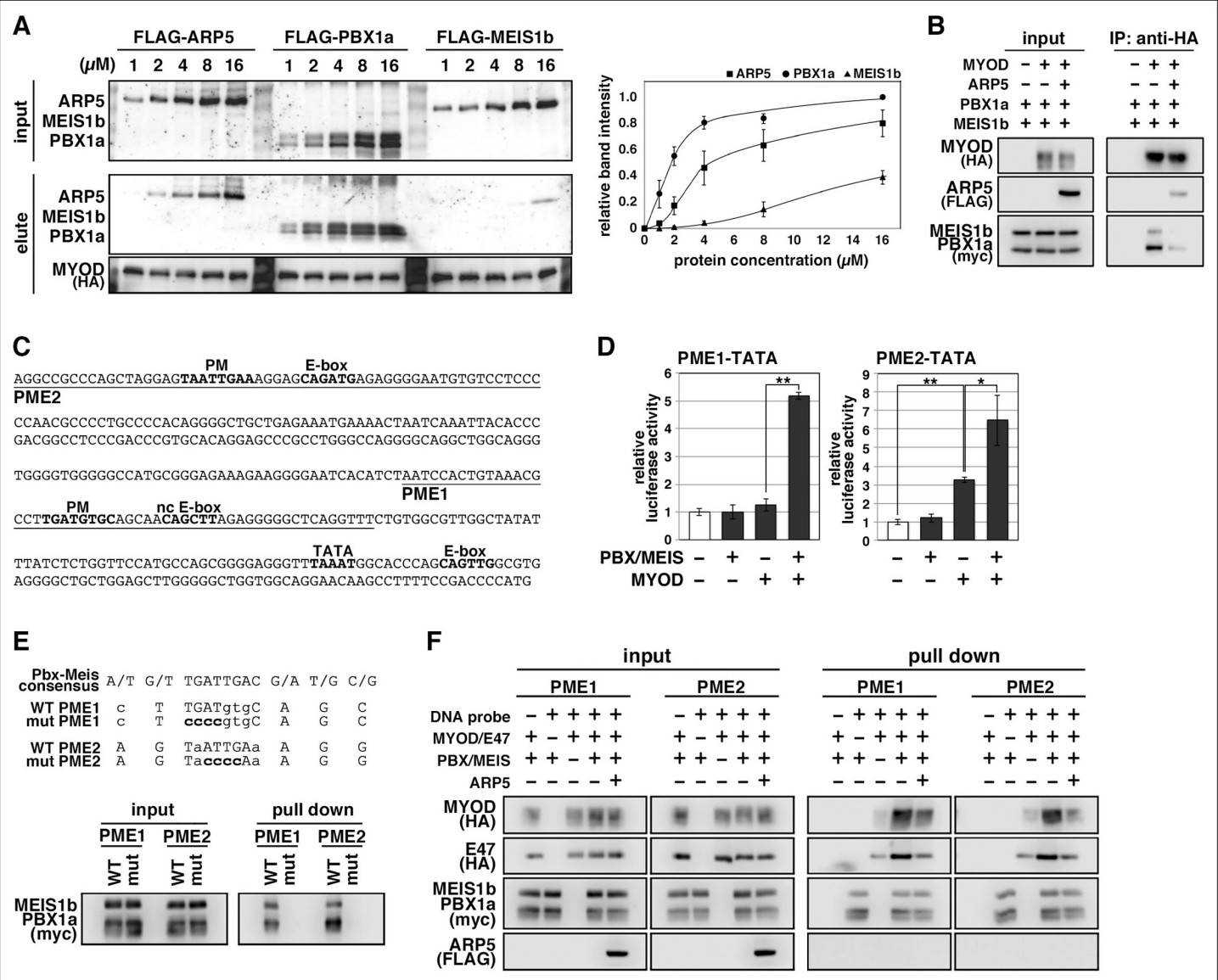

**Figure 5.** Actin-related protein 5 (Arp5) disturbs the interaction between MyoD and the Pbx1–Meis1 heterodimer. (**A**) Pull-down assay of MyoD-conjugated beads with purified Arp5, Pbx1a, and Meis1b proteins (left). Band intensities of eluted proteins were measured and plotted on a graph (right). (**B**) Co-immunoprecipitation assay between exogenous MyoD and Pbx1a–Meis1b. Co-incubation with Arp5 protein (lanes 3 and 6) diminished their interaction. (**C**) Sequence of the proximal promoter region of human *MYOG*. The core sequences of the indicated *cis*-regulatory elements are highlighted in bold. Pbx1–Meis1 heterodimer-binding motif/noncanonical E-box (PM/ncE)-containing regions (PME1 and PME2) are underlined. (**D**) PME1/2-TATA-controlled luciferase reporter assay in C2C12 cells. All statistical data are presented as the mean ± standard error of the mean (SEM). *p < 0.05, **p < 0.01 (Student's *t*-test). (**E**) Pull-down assay of PME1/2 DNA probes with Pbx1a–Meis1b protein. The consensus sequence of the PM motif is presented. The mutated nucleotides in the mut PME1/2 probes are highlighted in bold. (**F**) Pull-down assay of PME1/2 DNA probes with MyoD, E47, Pbx1a–Meis1b, and Arp5 proteins.

The online version of this article includes the following source data for figure 5:

**Source data 1.** Raw data used for the statistical analyses presented in *Figure 5*.

**Source data 2.** Original western blot data for *Figure 5A, B, E and F*.

cells (25.8%). Analysis of motif enrichment (AME) revealed several motif sequences that were more enriched in Arp5-KO peaks. Among them, CAGCTG matched the consensus sequence of the MyoD-binding site, and TGAC/GTCAT was similar to the previously reported Pbx–Meis complex-binding site TGATTnAT (*Blasi et al., 2017*; *Figure 7G*); 4156 of Arp5-KO peaks were either unique to Arp5-KO cells or had higher RPM peak scores relative to their cognate WT peaks. In addition to the TGAC/

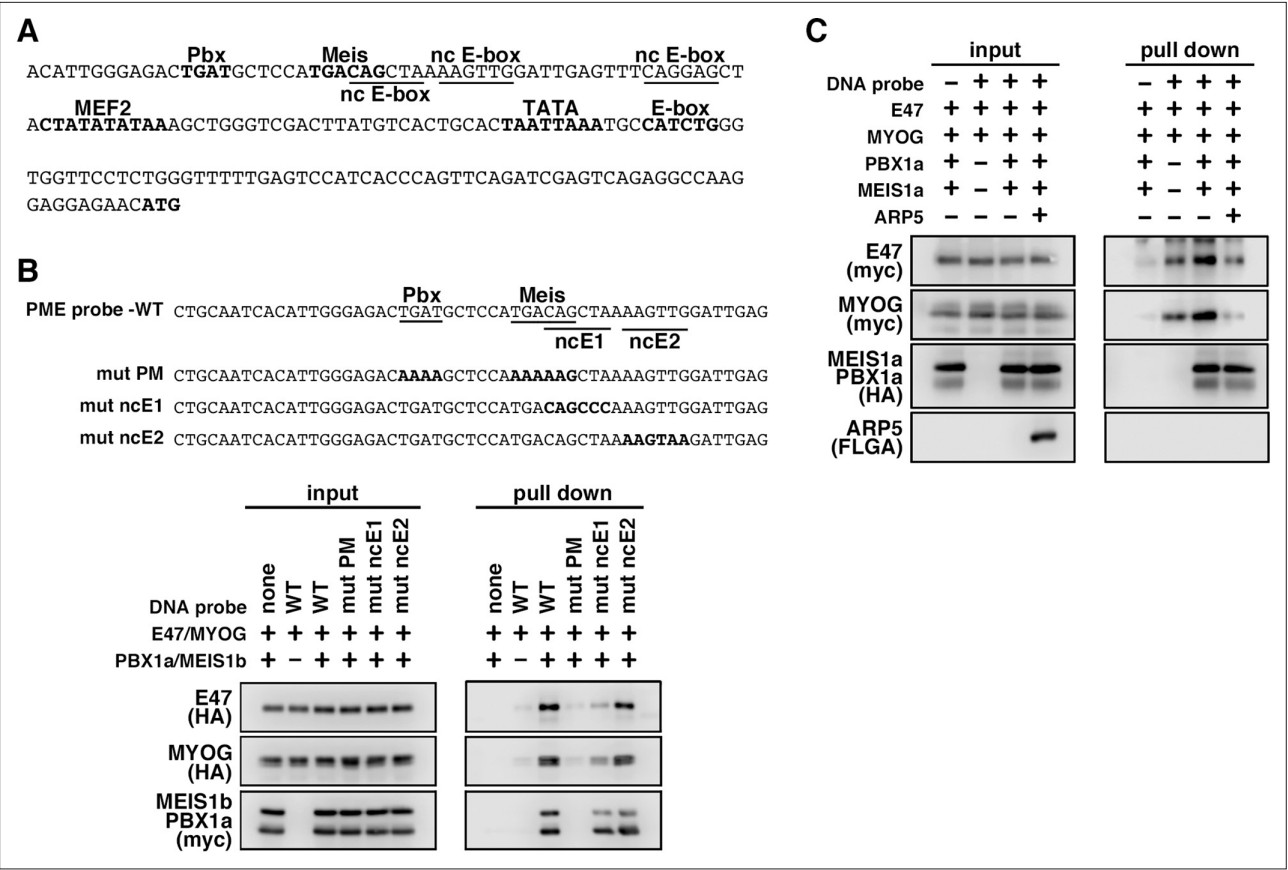

**Figure 6.** MyoG and the Pbx1–Meis1 heterodimer recognize the proximal promoter region of human myogenic regulatory factor 6 (MRF6). (**A**) Sequence of the proximal promoter region of human *MRF6*. The core sequences of the indicated *cis*-regulatory elements are highlighted in bold. The core sequences of putative noncanonical E-box motif (nc E-box) are underlined. (**B**) Pull-down assay of Mrf6 Pbx1–Meis1 heterodimer-binding motif/noncanonical E-box (PM/ncE)-containing region (PME) probes with MyoG, E47, and Pbx1a–Meis1b proteins. The mutated nucleotides in the mut PME probes (mut PM, mut ncE1, and mut ncE2) are highlighted in bold (top). The Pbx1–Meis1 heterodimer recognized the gapped Pbx–Meis-binding motif, while MyoG–E47 bound to the ncE1 site with the Pbx–Meis complex (bottom). (**C**) Pull-down assay of Mrf6 PME probes with MyoG, E47, Pbx1a–Meis1b, and Arp5 proteins.

The online version of this article includes the following source data for figure 6:

**Source data 1.** Original western blot data for *Figure 6B and C*.

GTCAT motif, two other Pbx–Meis-binding motifs were statistically significantly enriched in these KO-predominant peaks: TGAGTGACAG and TGACAG (*Figure 7H*). The former sequence almost matches with the PM motif (TGATTGACag), which is observed in the Myog and Mrf6 promoters (*Figures 5B and 6A*), and the latter was also previously reported as a Pbx–Meis-binding site (*Blasi et al., 2017*). Myocyte enhancer binding factor 2 (MEF2) is known to regulate the myogenic gene expression cooperatively with MyoD and MyoG (*Molkentin and Olson, 1996*). MEF2-binding motif CTAAAAATAG was also significantly enriched in the KO-predominant peaks (*Figure 7H*). These Pbx–Meis-binding motifs were also identified in the KO peaks whose peak scores were unchanged or decreased relative to their cognate WT peaks, but the statistical significance of their enrichment were comparatively low (*Figure 7H*). As shown in *Figures 5 and 6*, the Pbx–Meis complex recognizes the binding sites even when their sequences are rather different from those of the consensus Pbx–Meis-binding motif. Therefore, we investigated whether our ChIP-Seq peaks overlap with the Pbx ChIP-Seq peaks from a public ChIP-Seq database ChIP Atlas (*Figure 7I*). Enrichment analysis within ChIP Atlas demonstrated that a large percentage of the KO-predominant peaks predictably overlapped with the previously reported MyoD1 ChIP-Seq peaks. They also highly significantly overlapped with Pbx1/3 and Brg1 ChIP-Seq peaks more frequently than with MEF2 ChIP-Seq peaks. Thus, the knockout of Arp5 promotes recruitment of MyoD to its target loci shared by Pbx and Brg1.

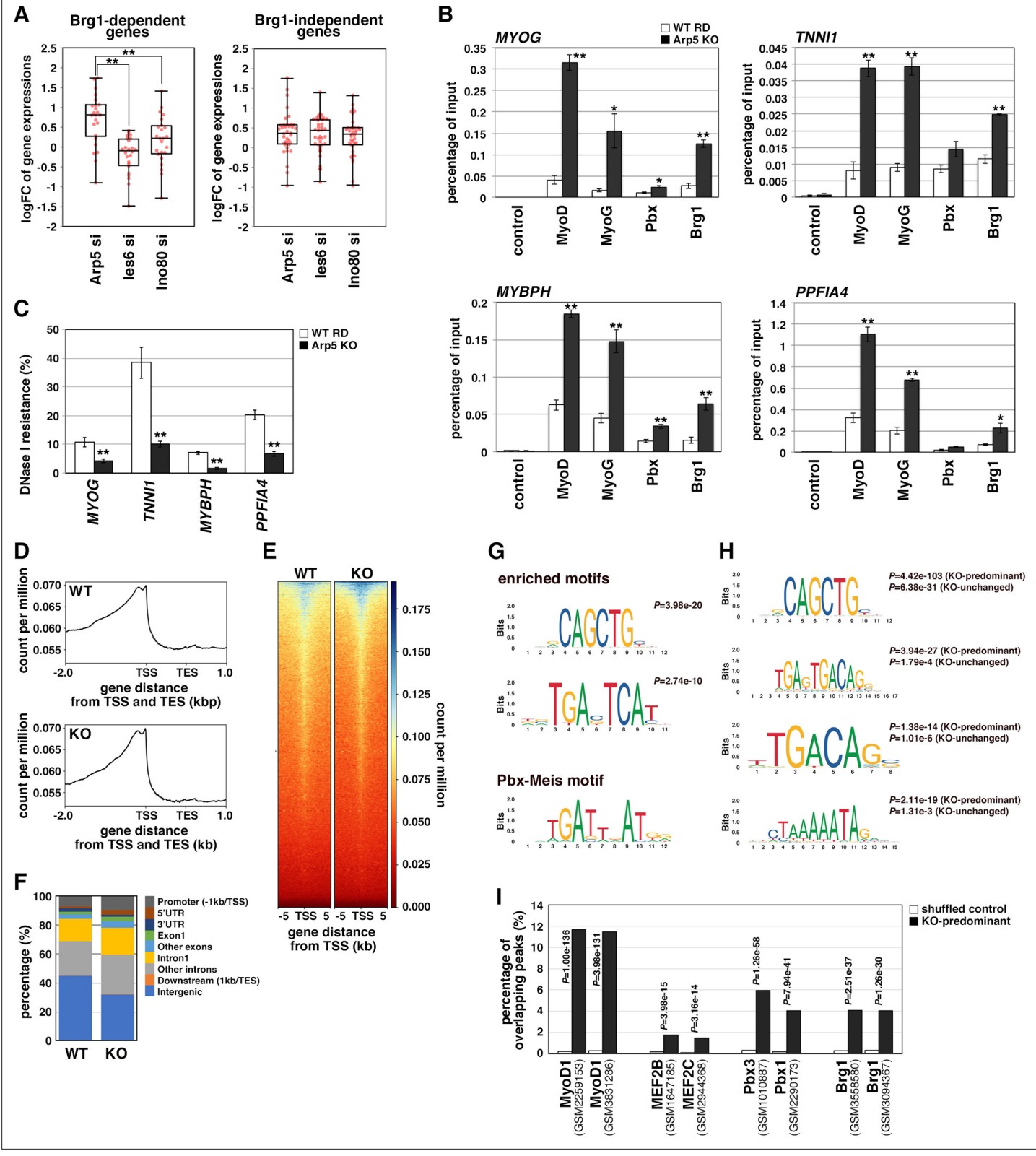

**Figure 7.** Actin-related protein 5 (Arp5) inhibits the recruitment of MyoD, MyoG, and Brg1-based switch/sucrose nonfermentable (SWI/SNF) to the enhancer region of myogenic genes. (**A**) Box-and-whisker plot of fold changes (log2) in the expression level of Brg1-dependent and Brg1-independent myogenic genes by *ACTR5*, *IES6*, and *INO80* knockdown. (**B**) Chromatin immunoprecipitation (ChIP) analysis using antibodies against MyoD, MyoG, Pbx, and Brg1 in wild-type (WT) and Arp5-knockout (KO) cells. Nonimmune IgG was used as a control. Enrichment efficiency of DNA fragments

*Figure 7 continued on next page*

*Figure 7 continued*

of *MYOG*, *TNNI1*, *MYBPH*, and *PPFIA4* enhancer loci was quantified by real-time polymerase chain reaction (PCR). (**C**) DNase I sensitivity assay of the enhancer loci in WT and Arp5-KO RD cells. Statistical data in (**A**), (**B**), and (**C**) are presented as the mean ± standard error of the mean (SEM). *p < 0.05, **p < 0.01 (Student's *t*-test). (**D**) Meta-gene profiles of MyoD ChIP-Seq in WT and Arp5-KO RD cells. TSS = transcription start site, TES = transcription end site. (**E**) Density heatmaps of MyoD ChIP-Seq reads around the TSS in WT and Arp5-KO RD cells. (**F**) Genomic distributions of MyoD-binding peaks in WT and Arp5-KO RD cells. (**G**) Enriched motifs in the Arp5-KO peaks compared to the WT peaks. Statistical analyses were performed by Fisher's exact test. The motif previously reported as a Pbx–Meis-binding site is shown in a lower panel. (**H**) Enriched motifs in the KO-predominant and KO-unchanged peaks. Statistical analyses were performed by Fisher's exact test. (**I**) Enrichment analysis of the KO-predominant peaks. Percent overlap between the KO-predominant peaks and MyoD, MEF2, Pbx, and Brg1 ChIP-Seq peaks from published database was calculated, respectively. Accession numbers of the used database were indicated. Statistical analyses were performed by Fisher's exact test using mononucleotide-shuffled sequences of the KO-predominant peaks as a control.

The online version of this article includes the following source data and figure supplement(s) for figure 7:

**Source data 1.** Raw data used for the statistical analyses presented in *Figure 7*.

**Figure supplement 1.** Putative enhancer regions recognized by Pbx1 and MyoD/MyoG in human *MYBPH*, *TNNI1*, and *PPFIA4*.

**Figure supplement 2.** Schematic representation of the role of Arp5 in myogenic gene expression.

## Discussion

In this study, we demonstrated a novel role of Arp5 in skeletal muscle differentiation. The H/C region of MyoD is critical for its chromatin remodeling activity. MyoD interacts with some epigenetic regulators, such as SWI/SNF, histone deacetylase, p300 histone acetyltransferase, and p300/CBP-associated factor (*Yuan et al., 1996*; *Puri et al., 1997*; *Lu et al., 2000*; *de la Serna et al., 2001*), but how the H/C region contributes to epigenetic regulation is unclear. The Pbx1–Meis1 heterodimer is so far the only identified partner of MyoD-binding directly to this region. Here, we reported Arp5 as another binding partner, which competes with the Pbx1–Meis1 heterodimer for binding to the H/C region. The Pbx1–Meis1 heterodimer is believed to recognize the PM motif in the enhancer region of myogenic genes and acts as a pioneering factor by marking this locus for the recruitment of MyoD and its co-factors, such as epigenetic regulators (*Cho et al., 2015*). Our findings clearly show that the Pbx1–Meis1 heterodimer directly binds to the PM motifs in the promoter region of *MYOG* and *MYF6* and augments the MyoD and MyoG binding to the canonical and noncanonical E-box close to the PM motifs. The ChIP and ChIP-Seq analysis reveal Pbx1 recruitment to the enhancer regions of some myogenic genes with MyoD. Arp5 inhibits the recruitment of MyoD and MyoG to Pbx1–Meis1-marked loci and, consequently, attenuates the recruitment of chromatin remodelers, such as Brg1–SWI/SNF. Therefore, low Arp5 expression in skeletal muscle tissues contributes to an increase in the chromatin accessibility of MyoD/MyoG target loci and the maintenance of high transcriptional activity of myogenic genes (*Figure 7—figure supplement 2*).

Arp5 is a well-known subunit of INO80 and plays an essential role in its ATPase activity and nucleosome sliding (*Yao et al., 2015*). In HeLa cells knocked down for each subunit of INO80, such as Ino80, Arp8, Ies2, and Ies6, the expression of many genes was altered, and these genes were enriched in several functional pathways such as the p53 signaling pathway, cell cycle, focal adhesion, and extracellular matrix–receptor interaction (*Cao et al., 2015*). These profiling data are in good agreement with our results using Ino80-si RD cells (*Figure 2—figure supplement 1A-E*). In osteogenic differentiation of mesenchymal stem cells, INO80 interacts with the WD repeat domain 5 protein and regulates osteogenic marker expression (*Zhou et al., 2016*). When 11 kinds of INO80 subunits are knocked down in mesenchymal stem cells, all the knockdown cells show a reduction in calcium deposition. Thus, each subunit seems indispensable for the function of INO80. In contrast, we observed many muscle-related genes to be upregulated only in Arp5-si cells but not in Ies6- and Ino80-si cells. Arp5 inhibited the activation of the MyoG promoter by MyoD, even in the absence of Ino80. Besides, Arp5 alone interrupted MyoD/MyoG–Pbx1–Meis1 complex formation. These results suggest that Arp5 regulates the activities of MyoD and MyoG independently of INO80. Recently, *Schutt et al., 2020*, reported that the INO80 complex is an interacting partner of Linc-MYH, a muscle-specific long intergenic noncoding RNA (*Schutt et al., 2020*). Together, they cooperatively regulate muscle stem cell proliferation through the YY1 and/or p53 signaling pathways. In 293T cells (which are non-muscle cells), knockdown of Ino80 also prolongs the progression from cell cycle phase G2/M to G1 by regulating the p53 signaling pathway (*Cao et al., 2015*). In contrast to the myogenic genes, expression of

genes related to the p53 signaling pathway was significantly altered in Ino80-si and Ies6-si RD cells (*Figure 2—figure supplement 1C, E*). Particularly, p21 (*CDKN1A*) expression was markedly increased compared to Arp5-si cells (*Figure 2—source data 3*), suggesting that the INO80 complex is involved in the regulation of cell cycle progression in a manner different from the regulation of myogenic genes by Arp5. We previously demonstrated the INO80-independent role of Arp5 in regulating the phenotypic plasticity of SMCs through direct interaction with myocardin (*Morita and Hayashi, 2014*). *Yao et al., 2016*, reported that the total abundance of Arp5 in cells is approximately eightfold of that as a component of INO80. Thus, Arp5 seems to have multiple functions besides being a subunit of INO80.

RMS cells are useful for analyzing the antimyogenic function of Arp5 because they highly express Arp5 and MRFs simultaneously. MyoD and MyoG are potent markers of RMS and can induce terminal myogenic differentiation, while their activities are post-transcriptionally dysregulated in RMS (*Keller and Guttridge, 2013*). Our results provide insight into the post-transcriptional regulation of MyoD and MyoG: abundant Arp5 in RMS induces MyoD and MyoG dysfunction via direct interaction. Arp5 deletion in RD cells leads to a loss of their tumorigenicity in nude mice due to the enhancement of myogenic differentiation. Arp5 is also identified as a T-cell acute lymphocytic leukemia and sarcoma antigen (*Lee et al., 2003*), although its contribution to carcinogenesis and sarcomagenesis is unclear. In Arp5-si RD cells, the expression of several cancer-associated genes decreased, including those reported to be mutated and abnormally expressed in RMS (*Figure 2—figure supplement 1F*). Survival data from The Cancer Genome Atlas reveal that high Arp5 expression is unfavorable for the prognosis of liver and renal carcinoma (data not shown). Thus, Arp5 likely has oncogenic properties, in addition to its inhibitory role in myogenesis.

Here, we reported that Arp5 expression is remarkably low in both human and mouse skeletal muscle. In contrast, RMS shows increased Arp5 expression with dysregulation of myogenic differentiation compared with tumor-adjacent skeletal muscles. *Haldar et al., 2007*, reported a synovial sarcoma mouse model in which the ectopic expression of a chimeric SYT-SSX fusion gene was driven from the *Myf5* promoter. In the tumors, Arp5 expression was approximately fivefold upregulated, while several myogenic markers were downregulated. Thus, Arp5 expression seems to be regulated according to the differentiation stage of skeletal muscle lineage cells. Alternative splicing coupled to nonsense-mediated messenger RNA (mRNA) decay (AS-NMD) is important for keeping the expression of Arp5 low in differentiated SMCs (*Morita and Hayashi, 2014*). AS-NMD contributes to post-transcriptional fine-tuning of broad gene expression (*Nasif et al., 2018*); furthermore, depletion of the splicing factor U2AF35 increased *Arp5* mRNA levels in HEK293 cells (*Kralovicova et al., 2015*). Tissue-specific alternative splicing is most frequently observed in skeletal muscle, and these splicing events are critical for proper skeletal muscle development and function (*Nakka et al., 2018*). These facts raise the possibility that AS-NMD contributes to the regulation of Arp5 expression in skeletal muscle and RMS.

This study reported a novel function of Arp5 in myogenic differentiation and tumorigenesis. Arp5 is a novel modulator of MRFs in skeletal muscle differentiation. Our data also cultivated a better understanding of chromatin remodeling events mediated by MRFs and Pbx–Meis in myogenic gene expression. The major limitations of this study are the lack of data on the change in Arp5 expression and its relevance to MRF activation during skeletal muscle development in vivo. Although in vivo experiments were performed using the AAV6 vector, further studies are needed to fully elucidate the mechanism and significance of low Arp5 expression in muscle tissues. It will be also interesting to investigate whether transcriptional and post-transcriptional regulation of Arp5 contributes to physiological and pathological skeletal muscle dysfunction.

## Materials and methods

### Cell lines, cell cultures, treatment, and transfections

Human RMS cells RD, C3H mouse muscle myoblasts C2C12, C3H mouse embryo cells 10T1/2, and human embryonic kidney cells HEK293T were obtained from ATCC. All cell lines have been tested negative for *Mycoplasma*. RD, 10T1/2, and HEK293T cells were cultured in Dulbecco's modified Eagle's medium (DMEM) supplemented with 10% FCS. C2C12 cells and human primary skeletal myoblasts (Thermo Fisher Scientific) were cultured in DMEM supplemented with 20% FCS (growth medium for myoblasts). Mouse primary myoblasts were isolated from the hind limbs of 3-week-old C57BL/6j mice as follows: The extracted hind limb muscle tissues were minced and incubated in a digestion solution

(260U type I collagenase, 3000 PU dispase II, and 10 mM $CaCl_2$ in 1 mL of Hanks' Balanced Salt solution [HBSS]) for 60 min at 37°C. The dispersed cells were washed twice with HBSS and plated on a type I collagen-coated culture dish with DMEM. After 30 min of incubation, the culture medium was removed, and the attached cells were cultured with the myoblast growth medium. The differentiated myoblasts were incubated in differentiation medium (DMEM supplemented with 2% house serum) for 1–3 days.

For myogenic transdifferentiation of the mouse embryo cell line, 10T1/2 cells were treated with 3 µM 5-azacitidine for 24 hr and then cultured in DMEM supplemented with 10% FCS for 3–7 days.

To establish Arp5-KO cell lines, RD cells were transfected with the all-in-one Cas9/gRNA plasmid pSpCas9 BB-2A-GFP (PX458; Addgene; gRNA target sequence, ccgttccgcgacgcccgtgccgc). One day after transfection, green fluorescent protein (GFP)-positive cells were sorted and individually cultured. Gene KO in the isolated clones was validated by DNA sequencing and western blotting.

In knockdown experiments, cells were transfected with predesigned siRNAs using Lipofectamine RNAiMAX (Thermo Fisher Scientific). The siRNA sequences are listed in *Table 1*. In overexpression experiments, cells were transfected with pCAGGS expression vectors using Lipofectamine 3000 (Thermo Fisher Scientific). The coding sequences of human *ACTR5*, *MYOD1*, *MYOG*, *E47*, *PBX1A*, and *MEIS1b* were inserted into vectors with FLAG-, HA-, and Myc-tag sequences. For the intramuscular expression of exogenous Arp5, the adeno-associated virus pseudotype 6 (AAV6) expression vector encoding *ACTR5* was constructed using AAVpro Helper Free System (AAV6) (TAKARA BIO). Finally, titration of AAV6 particles was performed using the AAVpro titration kit (TAKARA BIO).

## Animal studies

All animal experiments were conducted in accordance with the guidelines for animal experiments specified by the Wakayama Medical University, Japan, and Osaka University School of Medicine, Japan.

For Arp5 overexpression in mouse skeletal muscle tissues, we injected $1 \times 10^9$ gv/µL of control-AAV6 (empty vector) or Arp5-AAV6 vector injected into the hind limbs of 5-day-old C57BL/6j mice. Five weeks after injection, the hind limb muscles were extracted, fixed in 10% formalin, embedded in paraffin, and then cut into 5-µm-thick sections. The sections were stained with hematoxylin and eosin and observed.

In xenograft experiments, $2.5 \times 10^7$ cells were suspended in 1 mL of an EHS-gel basement membrane matrix (FUJIFILM Wako Pure Chemical Corporation) diluted to 1:1 with phosphate-buffered saline (PBS), and 200 µL of the suspension was subcutaneously inoculated into both sides of the flank of 3-week-old nude mice. The tumor size was measured every week using a pair of calipers, and the tumor volume was estimated as volume = $1/2$(length $\times$ width$^2$). After 6 weeks of inoculation, the mice were sacrificed, and xenograft tumors were extracted to obtain total RNA.

## Immunocytochemistry

Cells were cultured on coverslips and fixed with 10% formaldehyde solution. Fixed cells were incubated in blocking solution (0.1% Triton X-100, 0.2% bovine serum albumin, and 10% normal goat serum in PBS) for 30 min at 37°C and further incubated in a primary antibody solution (1:100 dilution of antibodies in Can Get Signal immunostaining reagent [TOYOBO]) for 2 hr. Next, the cells were washed twice with PBS and incubated in a secondary antibody solution (1:400 dilution of Alexa 488-conjugated secondary antibody [Thermo Fisher Scientific] and 1:1000 dilution of Hoechst 33342 [Thermo Fisher Scientific]) in the blocking solution for 1 hr. The cells were again washed twice, mounted on glass slides with Fluoromount (Diagnostic BioSystems), and observed under an all-in-one fluorescence microscope (BZ-9000; Keyence).

## Western blotting

Western blotting was performed, as previously described (*Morita and Hayashi, 2018*). Briefly, total proteins were extracted from cells with 2% sodium dodecyl sulfate (SDS) sample buffer. The proteins were electrophoretically separated using 10% polyacrylamide gels and then transferred to polyvinylidene difluoride membranes. For western blotting, anti-Arp5 (Proteintech), anti-MyoG (Santa Cruz Biotechnology, Inc), anti-MyoD (Santa Cruz Biotechnology, Inc), anti-MYF6 (Santa Cruz Biotechnology, Inc), anti-MHC (Developmental Studies Hybridoma Bank), anti-glyceraldehyde 3-phosphate

**Table 1.** List of PCR primer sequences and short interfering RNA (siRNA) sequences used in this study.

**Real-time RT-PCR**

| Gene name | Species | Forward primer sequence (5'→3') | Anti-sense primer sequence (5'→3') |
|---|---|---|---|
| 18srRNA | mouse/human | gtaacccgttgaaccccatt | cccatccaatcggtagtagcg |
| 28srRNA | mouse/human | gcgacctcagatcagacgtg | gggtcttccgtacgccacat |
| rpl13a | mouse/human | tgccgaagatggcggagg | cacagcgtacgaccaccacct |
| actr5 | mouse/human | agcaagccagagacccctga | agcctttgggtacctgtccag |
| myod1 | mouse | ggatggtgtccctggttcttc | cctctggaagaacggcttcg |
| myog | mouse | ggcaatggcactggagttcg | gcacacccagcctgacagac |
| myf6 | mouse | ccagtggccaagtgtttcg | cgctgaagactgctggagg |
| myh1 | mouse | gtcaacaagctgcgggtgaa | ggtcactttcctgcacttggatc |
| myh2 | mouse | ggctgtcccgatgctgtg | cacacaggcgcatgaccaa |
| myh3 | mouse | ctccagcagcgtagagagcg | ctagttgacgactcagctcaccc |
| myh4 | mouse | gtgaagagccgagaggttcacac | ctcctgtcacctctcaacagaaagatg |
| acta1 | mouse | gcactcgcgtctgcgttc | cctgcaaccacagcacgatt |
| tnni1 | mouse | cgacctcccagtagaggttggc | gaaagataggtgagtggggctgg |
| myod1 | human | cccgcgctccaactgctc | cggtggagatgcgctccac |
| myog | human | ggcagtggcactggagttcag | gtgatgctgtccacgatgga |
| myf6 | human | gatttcctgcgcacctgca | cgaaggctactcgaggctgacg |
| cdh15 | human | ctggacatcgccgacttcatc | gagggctgtgtcgtaaggcg |
| mybph | human | aggcatctgtggactgccg | cctcatcacagcctcctccc |
| mylpf | human | gaggatgtgatcaccggagcc | tggtcagcagctcctccagg |
| tnni1 | human | ccatgtctggcatggaaggc | aggagctcagagcgcagcac |
| tnni2 | human | ggacacagagaaggagcggg | cgagtggcctaggactcggactc |
| tnnt2 | human | gcaggagaagttcaagcagcaga | ccggtgactttagccttcccg |
| tnnt3 | human | agaccctgcaccagctggag | cggctcctgagcgtggtgat |
| tnnc2 | human | gacggcgacaagaacaacga | ggtagaggcgactgtccactcc |
| myh3 | human | cgagacttcacctccagcagg | ctgtcctgctccagaagggc |
| myl1 | human | acatcatgtctatctgaatggagctctc | ctggagagtttgtcatgggtgtg |
| myl4 | human | catcatgtcagggtgaagcagagtc | catctcagctcacccagccg |
| hrc | human | cggtctgcgctccaggaag | ccagcatgtctgccagggc |
| atp2a1 | human | tgcaagtcsttgggctcga | ctgtgacacgggctcagagatg |
| ppfia4 | human | gccaaagaagatcatgcctgaag | gccatggctagtcccggaag |
| cdkn1a | human | gctctgctgcaggggacag | gaaatctgtcatgctggtctgcc |

**ChIP assay and DNase I sensitivity assay**

| Gene name | Species | Forward primer sequence (5'→3') | Anti-sense primer sequence (5'→3') |
|---|---|---|---|
| myog | human | ggccatgcgggagaaagaag | cgctggcatggaaccagag |
| tnni1 | human | gggaagacagatggtctgggc | ggcctatcacctcagaggctg |
| mybph | human | cggagctgatcgggtttcag | gcttcaggccacctgcct |

*Table 1 continued on next page*

*Table 1 continued*

**Real-time RT-PCR**

| Gene name | Species | Forward primer sequence (5'→3') | Anti-sense primer sequence (5'→3') |
|---|---|---|---|
| ppfia4 | human | gcgtgcctggctccagttac | cagggctgcagctgtcattg |

**Knockdown assay**

| Gene name | Species | Target sequence (5'→3') |
|---|---|---|
| control | – | MISSION siRNA Universal Negative control (Sigma) |
| actr5 | mouse | acagatggaccagtttcac |
| actr5 | human | cctggcatgaaagccagaa |
| ino80c (ies6) | human | actgcggttcagcaccatt |
| ino80 | human | agcagctgccctacgggca |

dehydrogenase (GAPDH) (Thermo Fisher Scientific), anti-FLAG (Sigma-Aldrich), anti-HA (Roche Applied Science), and anti-Myc (Santa Cruz Biotechnology, Inc) antibodies were used as primary antibodies. The raw images of western blotting can be found in the source data files.

## Real-time RT-PCR

Total RNA was isolated and then reverse-transcribed using RNAiso Plus (TAKARA BIO) and the Prime-Script RT Reagent Kit with gDNA Eraser (TAKARA BIO). Real-time RT-PCR was performed using the THUNDERBIRD SYBR qPCT Mix (TOYOBO) on the Mic real-time PCR cycler (Bio Molecular System). Nucleotide sequences of the primer sets used in this study are listed in *Table 1*. To detect the expression of the endogenous *Myog* gene as distinct from the exogenous 1, a reverse primer for real-time RT-PCR was designed for the 3' UTR of *Myog* gene that is not included in the exogenous *Myog* mRNA.

## Reporter promoter analysis

To examine the promoter activity of human *MYOG*, the proximal promoter region (−299/−1) of *MYOG* was isolated by PCR and inserted into the pGL3-basic vector (Promega). The DNA fragments of PME1 and PME2 in the *MYOG* promoter region were also amplified and inserted into the pGL3-γ-actin-TATA vector. These reporter vectors were transfected into cells, together with the pSV-βGal vector (Promega) and the indicated gene expression vectors. After 2 days of transfection, luciferase activity was measured using the Luciferase Assay System (Promega), which was normalized to β-galactosidase activity.

## Co-immunoprecipitation assay

HEK293T cells were transfected with expression vectors to synthesize recombinant proteins. The next day, cells were lysed with Lysis buffer (0.5% NP-40, 10% glycerol, and protease inhibitor cocktail [Nacalai Tesque] in PBS) and gently sonicated. The lysate was centrifuged to remove cell debris and then incubated with Protein G Sepharose Fast Flow (Sigma-Aldrich) for 1 hr at 4°C to remove nonspecifically bound proteins. After centrifugation, the cell lysate proteins were mixed in the indicated combinations and incubated with the ANTI-FLAG M2 Affinity Gel (Sigma-Aldrich) or the Anti-HA Affinity Matrix from rat immunoglobulin G (IgG)$_1$ (Sigma-Aldrich) for 2 hr at 4°C. The beads were washed thrice with 0.5% NP-40 in PBS, and binding proteins were eluted using SDS sample buffer.

For detecting the interaction between endogenous Arp5 and MyoD, RD cells were lysed with the Lysis buffer and gently sonicated. The lysate was centrifuged and then incubated with Protein G Sepharose Fast Flow resin for 1 hr at 4°C to remove cell debris and nonspecifically bound proteins. The cell extract was incubated with anti-Arp5 or anti-MyoD antibody for 4 hr at 4°C and again with Protein G Sepharose Fast Flow resin for 2 hr. This procedure was repeated with IgG obtained from non-immunized rabbits and mice used as controls. After the incubation, the beads were washed thrice

with Wash buffer (1% Triton X-100, 0.1% sodium deoxycholate, 150 mM KCl, and 20 mM Tris-HCl [pH 6.8]), and bound proteins were eluted using SDS sample buffer.

To determine the relative binding affinity of MyoD to Arp5, Pbx1a, and Meis1b, we partially purified these proteins. HEK293T cells transfected with an expression vector to synthesize recombinant FLAG-ARP5, FLAG-PBX1a, or FLAG-MEIS1b protein were lysed with the Lysis buffer. After removal of cell debris and nonspecifically bound proteins, the cell lysates were incubated with the ANTI-FLAG M2 Affinity Gel for 2 hr at 4°C. The beads were washed thrice with the Wash buffer, and binding proteins were eluted with 3× FLAG Peptide (Sigma-Aldrich) according to the manufacturer's instructions. Exogenous HA-MYOD protein was similarly purified from transfected HEK293T cells using Anti-HA Affinity Matrix but not eluted. The resulting MYOD-binding beads were incubated with the indicated amounts of purified proteins in the Lysis buffer for 6 hr at 4°C. The beads were washed thrice with the Wash buffer, and bound proteins were eluted using SDS sample buffer. Eluted proteins were analyzed by western blotting, and band intensities were quantified using the ImageJ software (*Schneider et al., 2012*).

## Protein–DNA pull-down assay

DNA probes for the protein–DNA pull-down assay were generated by PCR using 5′-biotinylated primers and then conjugated to Dynabeads M-280 Streptavidin (Thermo Fisher Scientific). The recombinant proteins synthesized in HEK293T cells were extracted and incubated with the DNA-probe-conjugated beads in 0.5% NP-40, 10% glycerol, and protease inhibitor cocktail (Nacalai Tesque) in PBS for 2 hr at 4°C. The beads were washed thrice with 0.5% NP-40 in PBS, and binding proteins were eluted using SDS sample buffer.

## ChIP assay

ChIP assays were performed in WT and Arp5-KO RD cells using the SimpleChIP Plus Enzymatic Chromatin IP Kit (Cell Signaling Technology) according to the manufacturer's instructions. Anti-MyoD, anti-MyoG, anti-Pbx1/2/3/4, and anti-Brg1 antibodies (all from Santa Cruz Biotechnology, Inc) were used for immunoprecipitation. For control experiments, IgG from non-immunized rabbits and mice was used.

## ChIP-Seq assay

ChIP-seq was prepared using $3 \times 10^7$ cells each of WT and Arp5-KO RD cells, with anti-MyoD antibody. ChIP libraries were constructed using NEBNext ChIP-Seq Library Prep Master Mix Set for Illumina (Illumina) and sequenced using 150 bp paired-end sequencing on a NovaSeq 6000 (Illumina). The sequence data were checked for quality using FastQC, then mapped onto the GRCh38 reference using BWA. Peak calling was performed using MACS2 with the input DNA sequence data as a control. Meta-gene profiles and density heatmaps were generated using deeptools computeMatrix, plotProfile, and plotHeatmap. AME within MEME suits (*McLeay and Bailey, 2010*) was used to search for enriched motifs in the Arp5-KO peaks relative to the WT peaks. For AME of the KO-predominant peaks, shuffled input sequence (word size = 1) was used as a control. To search for enriched transcription factors colocalized to KO-predominant peaks, enrichment analysis within ChIP Atlas database (*Oki et al., 2018*) was performed with shuffled input sequences as a control (permutation times × 10).

## DNase I sensitivity assay

DNase I sensitivity assays were performed, as previously reported with slight modifications (*Gerber et al., 1997*). Briefly, WT and Arp5-KO RD cells were suspended in 0.5% NP-40, 10 mM NaCl, 5 mM MgCl₂, and Tris-HCl at pH 7.4 for 10 min at 4°C to isolate the nuclei. After centrifugation, the pelleted nuclei were resuspended in DNase I buffer (10 mM NaCl, 6 mM MgCl₂, 1 mM CaCl₂, and 40 mM Tris-HCl at pH 8.0) and then treated with 1.5 U/50 mL of DNase I (Thermo Fisher Scientific) for 10 min at 37°C. The reaction was stopped by adding an equal volume of 40 mM ethylenediaminetetraacetic acid (EDTA). The digested nuclei were collected by centrifugation, suspended in 50 mM NaOH, and heated for 10 min at 100°C to extract DNA. The resulting solution was diluted 100 times with TE (1 mM EDTA, 10 mM Tris-HCl at pH 8.0). The digestion efficiency of the focused enhancer loci of myogenic genes was calculated by comparing the amount of intact enhancer fragments between DNase-I-treated and DNase-I-untreated samples using real-time PCR.

## DNA microarray

Total RNAs were isolated from RD cells transfected with control, Arp5, Ies6, or Ino80 siRNA for 2 days, using NucleoSpin RNA Plus (TAKARA BIO). RNAs from three independent experiments (biological replicates) were mixed in equal amounts and used as samples for subsequent microarray analysis. mRNAs were reverse-transcribed, and Cy3-labeled complementary RNAs (cRNAs) were synthesized using the Low Input Quick Amp Labeling Kit (Agilent Technologies). The cRNAs were hybridized on a SurePrint G3 Mouse Gene Expression 8 × 60 K Microarray (Agilent Technologies), and fluorescence signals were detected using the SureScan Microarray Scanner (Agilent Technologies). The fluorescence intensity was quantified using Feature Extraction Software (Agilent Technologies).

## Statistics and reproducibility

All statistical data were generated from experiments independently repeated at least three times, and values were expressed as the mean ± standard error of the mean (SEM). Data were assessed using Student's $t$-test, and $*p < 0.05$ and $**p < 0.01$ were considered statistically significant. The raw data for statistical analysis can be found in the source data files. For non-quantitative data such as western blotting, all experiments were independently replicated three times, and representative images are shown in the figure.

## Acknowledgements

We thank all the staff in the Laboratory Animal Center, Wakayama Medical University, and the Center for Medical Research and Education of Osaka University Graduate School of Medicine for their technical assistance. We also thank Enago (https://www.enago.jp) for the English language review. This work was supported by JSPS KAKENHI Grant Number 15K07076 (to T Morita), 18K06913 (to T Morita), and 19K07351 (to K Hayashi).

---

# Additional information

### Funding

| Funder | Grant reference number | Author |
|---|---|---|
| Japan Society for the Promotion of Science | KAKENHI 15K07076 | Tsuyoshi Morita |
| Japan Society for the Promotion of Science | KAKENHI 18K06913 | Tsuyoshi Morita |
| Japan Society for the Promotion of Science | KAKENHI 19K07351 | Ken'ichiro Hayashi |

The funders had no role in study design, data collection and interpretation, or the decision to submit the work for publication.

### Author contributions

Tsuyoshi Morita, Conceptualization, Formal analysis, Funding acquisition, Investigation, Methodology, Project administration, Resources, Supervision, Validation, Visualization, Writing - original draft, Writing - review and editing; Ken'ichiro Hayashi, Conceptualization, Methodology, Resources, Writing - original draft

### Author ORCIDs

Tsuyoshi Morita http://orcid.org/0000-0002-9022-298X

### Ethics

All animal experiments were conducted in accordance with the guidelines for animal experiments specified by the Wakayama Medical University, Japan, and Osaka University School of Medicine, Japan. The Protocols were approved by the Committee on the Ethics of Animal Experiments of the Wakayama Medical University (Permit Number: 900 ) and Osaka University School of Medicine (Permit Number: 28-018).

Decision letter and Author response
Decision letter https://doi.org/10.7554/eLife.77746.sa1
Author response https://doi.org/10.7554/eLife.77746.sa2

## Additional files

### Supplementary files
• Transparent reporting form

### Data availability

DNA microarray data have been deposited in the GEO database https://www.ncbi.nlm.nih.gov/geo/query/acc.cgi?acc=GSE169681 (accession no. GSE169681). ChIP-Seq data have been deposited in the DDBJ database https://ddbj.nig.ac.jp/resource/bioproject/PRJDB13012 (Run accession no. DRR345782-DRR345785). All data generated or analysed during this study are included in the manuscript and supporting files; Source Data files have been provided for Figures 1, 2, 3, 4, 5, 6 and 7.

The following datasets were generated:

| Author(s) | Year | Dataset title | Dataset URL | Database and Identifier |
|---|---|---|---|---|
| Morita T, Hayashi K | 2021 | Knockdown of Ino80 complex subunits Actr5, Ies6, and Ino80 in RD cells | https://www.ncbi.nlm.nih.gov/geo/query/acc.cgi?acc=GSE169681 | NCBI Gene Expression Omnibus, GSE169681 |
| Morita T | 2022 | Comparison of MyoD binding sites in WT and Arp5-KO RD cells [ChIP-seq] | https://ddbj.nig.ac.jp/resource/bioproject/PRJDB13012 | DNA Data Bank of Japan, PRJDB13012 |

The following previously published dataset was used:

| Author(s) | Year | Dataset title | Dataset URL | Database and Identifier |
|---|---|---|---|---|
| Sarver A | 2011 | Endothelial and Alveolar Rhabdomyosarcoma mRNA expression | https://www.ncbi.nlm.nih.gov/geo/query/acc.cgi?acc=GSE28511 | NCBI Gene Expression Omnibus, GSE28511 |

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
