## [Editor Report]

This study describes a novel mechanism by which ARP5 interacts with the master transcription factor MYOD to antagonize its function and thereby the myogenic gene expression program. ARP5 is up-regulated in rhabdomyosarcoma cells and appears to inhibit MYOD function as its depletion partially restores the myogenic program in the tumour cells. ARP5 is therefore a novel regulator of myogenesis in both normal and pathological contexts.

---

## [Decision Letter]

**Decision letter after peer review:**

[Editors’ note: the authors submitted for reconsideration following the decision after peer review. What follows is the decision letter after the first round of review.]

Thank you for submitting the paper "Actin-related protein 5 functions as a novel modulator of MyoD and MyoG in skeletal muscle and in rhabdomyosarcoma" for consideration at *eLife*. Your submission has been reviewed by three peer reviewers, one of whom is a member of our Board of Reviewing Editors, and the evaluation has been overseen by a Senior Editor . Although the work is of interest, we regret to inform you that the findings at this stage are too preliminary for further consideration at *eLife*.

The referees felt that this paper had potential, but was currently premature for publication in *eLife*. Specifically, the evidence for direct ARP5-MYOD interaction is weak and evidence of its competitive nature with PBX1-MYOD is also weak. No information on the relative affinities of ARP5-MYOD versus PBX1-MYOD interactions was provided. Evaluation of how APR5 impacts MYOD and BRG1 recruitment is limited to ChIP-PCR on some selected sites whereas it could have been evaluated on a more global level. To be suitable for *eLife*, the referees would have expected strong evidence for direct interactions either using recombinant proteins or at least IP or gel filtration with the endogenous factors and not simply ectopic expression in HEK cells. Similarly, MYOD and BRG1 ChIP-seq possibly also ATAC-seq should be performed on the ARP5 expressing and KO RD cells to evaluate the full effect of ARP5 on MYOD function BRG1 recruitment and chromatin remodeling.

We are however willing to consider a novel version of the study that will be treated as a new submission if all of the issues raised by the referees can be satisfactorily addressed. To be reconsidered, the authors have to provide more rigorous data that fully characterize the ARP5-MYOD1 interactions and quantify the competitive nature with the MYOD1-PBX1 interactions. Suggestions as to the experiments required are outlined in the detailed reviews below.

*Reviewer #1:*

The authors of this study provide evidence of a novel mechanism by which the ARP5 (ACTR5) protein regulates myogenic differentiation. The study reveals that ARP5 interacts with MYOD to antagonize its function and thereby the myogenic gene expression program. ARP5 interaction with MYOD1 regulates the PBX1/MEIS-MYOD interaction involved in recruitment of the BRG1-containing SWI/SNF chromatin remodeling complex required for gene activation. While the effect of ARP5 on myogenic differentiation is well described in cellular models and in mouse, the mechanisms involved require more detailed investigation and a genome-wide approach would provide a better understanding of the global impact of the ARP5-MYOD interaction on chromatin remodeling and gene expression.

In Figure 1C the authors should perform ARP5 immunoblots and/or RT-qPCR on the infected muscle tissues to assess ARP5 expression.

The authors show the consequences of ARP5 overexpression on C2C12 cells whereas the siRNA experiments seem to be performed in primary myoblasts. Why are the gain and loss of function experiments not all performed in both C2C12 and primary mouse myoblasts?

One important issue in this study is that ARP5 is a known subunit in the INO80 complex. The results of the silencing of ARP5 vs other INO80 subunits shows different effects on gene expression. However, in molecular terms is there any information on whether the ARP5 that interacts with MYOD to inhibit myogenesis is a free non-INO80 associated form. Can INO80 or other subunits of the complex be co-precipitated with MYOD. In this study, the authors show only co-immunoprecipitation of ectopically expressed proteins, not of the endogenous proteins. The authors should show co-IP of the endogenous proteins (MYOD, ARP5 INO80), however if there is an issue with suitable antibodies for IP, the authors should perform gel filtration experiments on RD cells and examine if there is free ARP5 and determine if MYOD co-elutes with either the free ARP5 or the INO80 associated form. It would also be appropriate to show an immunoblot of RD cells side by side with C2C12 cells for example to demonstrate its much higher expression in the RD cells.

The main hypothesis of the authors is that ARP5 competes with PBX1/MEIS for binding to the H/C domain of MYOD. The only experiment that directly addresses this issue is that shown on Figure 5A where ectopic expression of ARP5 leads to reduced PBX1/MEIS-MYOD-co-IP. This experiment provides no attempt at quantification of the relative affinities of binding of ARP5 vs PBX1/MEIS for MYOD. Also, can the authors explain why they used immobilized templates assays with the promoter DNA rather than gel shift experiments that would have allowed titration using increasing amounts of either ARP5-containing extracts or PBX1/MEIS-containing extracts. While asking for reconstitution of these interactions using purified recombinant proteins is beyond the scope of this study some attempts should be made to quantify the relative affinities of ARP5 and PBX1/MEIS for MYOD.

Lastly on the issue of protein-protein interactions this referee is rather intrigues by the very low stringency of the IP washes that are performed with PBS and NP40. Are these interactions so weak they cannot be observed under more stringent conditions such as 150-300 mM KCL for example.?

While the ChIP-qPCR experiments shown in Figure 7 clearly show increased MYOD and BRG1 recruitment to selected regions, it would have been much more informative to have examined MYOD1 and BRG1 recruitment by ChIP-seq in RD cells to investigate globally how they are affected by ARP5 KO. For example, a genome-wide analyses would have indicated if MYOD sites with adjacent PM motifs were preferentially affected. This type of analyses would be appropriate for publication in *ELife*.

*Reviewer #2:*

This study investigated the regulation of muscle differentiation by ARP5, which is a component of the INO80 chromatin remodeling complex. ARP5 was found to be expressed at a low level in muscle and over-expressed in rhabdomyosarcoma cells. Ectopic expression of ARP5 in muscle tissue caused atrophy and inhibited differentiation of cultured myoblasts. Depletion of ARP5 in rhabdomyosarcoma promoted differentiation and inhibited tumorigenicity. The regulation of muscle differentiation by ARP5 was found not to involve other INO80 components. Co-immunoprecipitations and DNA pull downs suggested that ARP5 competes with PBX1 binding to a conserved cysteine-rich region of MYOD and Myogenin (CR). PBX1 is considered a pioneer factor, required for MYOD and myogenin to bind within repressive chromatin structure at PBX1 heterodimeric/noncanonical E box motifs in a subset of muscle specific genes and thereby allow recruitment of the BRG1 component of the SWI/SNF chromatin remodeling complex. Gene expression profiling of rhabdomyosarcoma cells indicated a correlation between ARP5 regulated genes and previously identified BRG1 regulated genes. Furthermore, ARP5 depletion in rhabdomyosarcoma cells enhanced MYOD and Myogenin binding as well as BRG1 recruitment and chromatin accessibility on muscle specific loci. Thus, ARP5 negatively regulates muscle specific gene expression by affecting chromatin structure. This study adds to our understanding of the transcriptional regulation of muscle differentiation and how it can be compromised in muscle disease.

Strengths

The strengths of this study include the novelty of identifying ARP5 as a negative regulator of muscle through an INO80-independent mechanism and differentiation and significance to human disorders such as muscle atrophy and tumorigenicity of rhabdomyosarcoma cells. The data indicating that ARP5 is over-expressed in atrophied muscles and rhabdomyosarcoma are strong as are the effects of Arp5 on muscle differentiation and tumorigenicity. Several aspects of the proposed mechanism by which ARP5 inhibits muscle specific gene expression are supported by the findings. There is convincing data that ARP5 binds to the CR region of MYOD and Myogenin and that it disrupts MYOD and Myogenin binding to PBX1 at PBX1 heterodimer binding/noncanonical E box motifs. Chromatin immunoprecipitations are convincing in showing enhanced enrichment of MYOD/Myogenin binding and recruitment of BRG1 to muscle specific loci with evidence of effects on chromatin remodeling upon ARP5 knockdown.

Weaknesses

Protein-protein interactions were investigated in HEK-293 cells which have other cellular proteins. Therefore, the claim that the interaction between ARP5 with MYOD and Myogenin is direct is not substantiated. Experimental details and additional controls for some experiments are needed.

1. Co-immunoprecipitations were done in HEK293 cells with other cellular proteins, which could be mediating the interactions between MYOD/Myogenin and ARP5. Pulldowns with purified recombinant proteins isolated from bacteria or in vitro-translated proteins that donot have other mammalian factors would better support that the interaction is direct.

2. Figure 1C and D should show the expression level of ectopic ARP5.

3. Figure 4F: How was endogenous Myogenin distinguished from transfected Myogenin expression? The amount of ARP5 over-expression should be shown for this experiment.

4. Although there is a statement in Methods regarding the number of independent experiments for statistical analysis, it is not clear how many experiments, representative figures like westerns, and other non-quantitative data represent. This should be clearly stated.

5. What is the negative control in Figure 7B? In general, the manuscript would benefit from more experimental details in the Methods section and in figure legends.

6. The sentence on page 10, line 211 which states that ARP5 affects the expression of MYOD target genes in a BRG1-dependent manner is an overstatement. The data from the microarray is correlative and the ChIP data in Figure 7 indicate an association with effects on BRG1 recruitment. It is still possible that ARP5 can have repressive effects on muscle specific gene expression by additional mechanisms that do not involve BRG1.

7. A schematic model of the regulatory events would help summarize the findings.

*Reviewer #3:*

In this manuscript, Morita and Hayashi have examined the role for ARP5 in modulating MRF function in rhabdomyosarcoma. Using normal muscle, they show that overexpression of ARP5 leads to decreased expression of target genes and impaired differentiation. Using a rhabdomyosarcoma cell line, they find that loss of ARP5 increases the expression of muscle genes, and prevents tumor growth upon transplantation. The altered expression of muscle genes is independent of the Ino80 complex occurs through an interaction with the Cys-rich domain present in MyoD and Myogenin. Pulldown assays suggest that Arp5 blocks MyoD (or Myog) function at specific genes by preventing interactions with Pbx1/Meis. In the absence of the MyoD-Pbx/Meis interaction, SWI/SNF is not recruited to the muscle-specific loci. Based on these results, the authors conclude that Arp5 is a novel modulator of MRFs in skeletal muscle differentiation.

Strengths

This manuscript described the novel interaction between MyoD and ARP5 as a regulatory axis of myogenic gene expression. Overexpression studies in mice show smaller fiber CSA, point to ARP5 as a new player in preventing muscle differentiation. The mechanism of action is explored, and strongly support a role for Arp5 blocking the interaction between the Cys-Rich region of MyoD with Pbx-Meis, and therefore preventing recruitment to the myogenin promoter.

Weaknesses

The interpretation is complicated by the fact that the authors have worked in two different cell systems – myoblasts and rhabdomyosarcomas. For instance, overexpression is performed in C2C12 to block differentiation, but the authors have not looked at knockdown in C2C12 cells. The knockdown of ARP5 in rhabdomyosarcomas allows expression of muscle genes and prevent tumor growth in transplants. Does loss of ARP5 in C2C12 cells lead to premature differentiation. This would strengthen the argument that ARP5 is an important modulator of MRF function controlling differentiation. Furthermore, it would be important to show that the same effects are observed in primary myoblasts.

1. A recent paper in EMBO J looked at the role of Ino80 as an interactor of Linc-Myh in the control of myoblast proliferation. In that study, Ino80 was required for myoblast proliferation. Here the authors have excluded the possibility that ARP5 is acting through Ino80 using qPCR of myogenic target genes in RMS cells after siRNA knock-down. The effect of knocking down ARP5 and Ino80 should be examined in proliferating C2C12 cells to see if they are acting through different pathways. Would overexpression of ARP5 continue to block differentiation if Ino80 was knocked down?

2. The authors have made a link between ARP5 expression and rhabdomyosarcoma tumor growth based on the over expression of ARP5 in the RMS cell line. While these results are convincing, it would be important to show the general interest of these findings in RMS. Indeed, many different factors have been shown to prevent tumour growth when knocked-down in RMS cells. The authors should examine RNA-Seq data of human rhabdomyosarcomas from the GEO data (or others) to see whether ARP5 is regularly over expressed in rhabdomyosarcomas compared to myoblasts.

3. The authors should examine the knockdown of ARP5 in proliferating C2C12 cells to determine the importance of ARP5 to blocking the myogenic differentiation activity of MyoD in myoblasts.

4. It would be nice to see ChIP-seq results for MyoD with and without ARP5 knockdown to assess whether MyoD binding is specifically lost at Pbx-Meis binding sites.

5. The effects of ARP5 overexpression and knock-down should be confirmed in primary myoblasts.

6. In figure 5e, the panel showing the ARP5 western lots in the pull down are missing. These should be added to the image.

[Editors’ note: further revisions were suggested prior to acceptance, as described below.]

Thank you for resubmitting your work entitled "Actin-related protein 5 functions as a novel modulator of MyoD and MyoG in skeletal muscle and in rhabdomyosarcoma" for further consideration by *eLife*. Your revised article has been evaluated by Kathryn Cheah (Senior Editor) and a Reviewing Editor.

The manuscript has been improved but there are some remaining issues that need to be addressed, as outlined below:

*Reviewer #1:*

In this new version of the study, the authors have addressed many of the issues raised in the original version. The data showing that ARP5 can compete MYOD-MEIS/PBX binding and weakened MYOD interaction with DNA and chromatin has now been addressed in a much more convincing way. Specifically, the authors have performed MYOD ChIP-seq in the ARP5 KO cells. They claim that binding of MYOD is increased in absence of ARP5 and new sites, or sites with strongly increased occupancy are observed. However, the presentation of this data is sparse. The authors should show the normalized meta-profiles of MYOD binding to commonly occupied sites in each condition. They should show read density heat maps for total non-redundant sites in each condition to illustrate specific MYOD binding to sites only in the absence of ARP5. The total number of sites detected in each condition should be shown and their genomic distributions illustrated. DNA motif analyses should be shown on sites where no change is seen in absence of ARP5 compared to sites that show enhanced MYOD occupancy. Figure 7F is very difficult to interpret. The authors should find a much better and more quantitative method for integrating these different data sets.

A more quantitative analysis of the data will strengthen the authors conclusions.

*Reviewer #3:*

The authors have done extensive new experimentation to address my previous concerns. These experiments have solidified the authors conclusions that Arp5 competes with Pbx/Meis for binding to the CR domain of MyoD to regulate function.

Page 6, line 104-105 the authors state "These results indicate that Arp5 overexpression contributes to MRF dysfunction in RMS."

This is an over interpretation of the results to this point. Please change to "These results suggest that Arp5 overexpression may contribute to MRF dysfunction in RMS."

---

## [Author Response]

[Editors’ note: the authors resubmitted a revised version of the paper for consideration. What follows is the authors’ response to the first round of review.]

Reviewer #1:The authors of this study provide evidence of a novel mechanism by which the ARP5 (ACTR5) protein regulates myogenic differentiation. The study reveals that ARP5 interacts with MYOD to antagonize its function and thereby the myogenic gene expression program. ARP5 interaction with MYOD1 regulates the PBX1/MEIS-MYOD interaction involved in recruitment of the BRG1-containing SWI/SNF chromatin remodeling complex required for gene activation. While the effect of ARP5 on myogenic differentiation is well described in cellular models and in mouse, the mechanisms involved require more detailed investigation and a genome-wide approach would provide a better understanding of the global impact of the ARP5-MYOD interaction on chromatin remodeling and gene expression.In Figure 1C the authors should perform ARP5 immunoblots and/or RT-qPCR on the infected muscle tissues to assess ARP5 expression.

We performed real time RT-PCR to assess Arp5 expression on the infected muscle tissues (Figure 1C) and confirmed the excessive expression of Arp5 in these tissues.

The authors show the consequences of ARP5 overexpression on C2C12 cells whereas the siRNA experiments seem to be performed in primary myoblasts. Why are the gain and loss of function experiments not all performed in both C2C12 and primary mouse myoblasts?

Following reviewer’s comment, we showed the gain and loss of function experiments in both C2C12 and primary mouse myoblasts (Figure 1E-J). In both cell types, Arp5 played an inhibitory role in the expression of myogenic genes.

One important issue in this study is that ARP5 is a known subunit in the INO80 complex. The results of the silencing of ARP5 vs other INO80 subunits shows different effects on gene expression. However, in molecular terms is there any information on whether the ARP5 that interacts with MYOD to inhibit myogenesis is a free non-INO80 associated form. Can INO80 or other subunits of the complex be co-precipitated with MYOD. In this study, the authors show only co-immunoprecipitation of ectopically expressed proteins, not of the endogenous proteins. The authors should show co-IP of the endogenous proteins (MYOD, ARP5 INO80), however if there is an issue with suitable antibodies for IP, the authors should perform gel filtration experiments on RD cells and examine if there is free ARP5 and determine if MYOD co-elutes with either the free ARP5 or the INO80 associated form. It would also be appropriate to show an immunoblot of RD cells side by side with C2C12 cells for example to demonstrate its much higher expression in the RD cells.

We performed the co-immunoprecipitation between endogenous Arp5 and MyoD in RD cells (Figure 4B), indicating physiologically significant interaction between them. In our study, we have not ruled out the possibility of the direct or indirect interaction between MyoD and INO80. We have argued that INO80 is not necessary for Arp5 to interact with and inhibit MyoD. This idea is further supported by new data: Arp5 can repress the MyoD-induced activation of the myogenin promoter even in the absence of Ino80 (Figure 4F), and purified Arp5 binds directly to MyoD (Figure 5A). We also showed the abundant expression of Arp5 protein in RD cells compared to myoblast cells (Figure 2B).

The main hypothesis of the authors is that ARP5 competes with PBX1/MEIS for binding to the H/C domain of MYOD. The only experiment that directly addresses this issue is that shown on Figure 5A where ectopic expression of ARP5 leads to reduced PBX1/MEIS-MYOD-co-IP. This experiment provides no attempt at quantification of the relative affinities of binding of ARP5 vs PBX1/MEIS for MYOD. Also, can the authors explain why they used immobilized templates assays with the promoter DNA rather than gel shift experiments that would have allowed titration using increasing amounts of either ARP5-containing extracts or PBX1/MEIS-containing extracts. While asking for reconstitution of these interactions using purified recombinant proteins is beyond the scope of this study some attempts should be made to quantify the relative affinities of ARP5 and PBX1/MEIS for MYOD.

We quantified the relative binding affinities of MyoD to Arp5, Pbx1, and Meis1 using purified proteins (Figure 5A) and indicated that Arp5 binds to MyoD with sufficient affinity, although slightly lower than Pbx1. This was also supported by a co-immunoprecipitation assay of endogenous proteins (Figure 4B). We used the immobilized template assay (protein-DNA pull-down assay) rather than the gel shift assay to detect the interaction between DNA and transcription factors for the following reasons. In our study, up to four different proteins, MyoD, E47, Pbx1, and Meis1, bound to the one DNA probe. In such a case, our method is more suitable for detecting the amount of binding each protein, because the binding proteins can be detected individually by western blot analysis. Gel shift assay detects proteins bound to DNA probes only in a complex state. In addition, the binding between the noncanonical E-box and MyoD is very faint and may be separated during electrophoresis by gel shift assay. Our experiments clearly demonstrated that the faint interaction between MyoD/E47 and noncanonical E-box was markedly augmented by Pbx1/Meis1, while Pbx1/Meis1 constantly bound to the DNA probe. Arp5 reduced the augmented interaction of MyoD/E47 without affecting the DNA binding of Pbx1/Meis. Another advantage of this method is that it does not require the use of RI, although this is a technical issue.

Lastly on the issue of protein-protein interactions this referee is rather intrigues by the very low stringency of the IP washes that are performed with PBS and NP40. Are these interactions so weak they cannot be observed under more stringent conditions such as 150-300 mM KCL for example.?

In the newly added immunoprecipitation experiments, a more stringent wash buffer (1% Triton X-100, 0.1% DOC, 150 mM KCl) was used. Even under this condition, the endogenous interaction between Arp5 and MyoD was detected (Figure 4B).

While the ChIP-qPCR experiments shown in Figure 7 clearly show increased MYOD and BRG1 recruitment to selected regions, it would have been much more informative to have examined MYOD1 and BRG1 recruitment by ChIP-seq in RD cells to investigate globally how they are affected by ARP5 KO. For example, a genome-wide analyses would have indicated if MYOD sites with adjacent PM motifs were preferentially affected. This type of analyses would be appropriate for publication in ELife.

We performed ChIP-seq analysis using anti-MyoD antibody in WT and Arp5-KO RD cells and demonstrated that Pbx-Meis-binding motifs were significantly enriched in the peak sequences from Arp5-KO cells (Figures 7D and 7E). Furthermore, the Arp5-KO peaks significantly overlapped with the ChIP-seq peaks of Pbx1 and Brg1 obtained from public database (Figure 7F).

Reviewer #2:[…]1. Co-immunoprecipitations were done in HEK293 cells with other cellular proteins, which could be mediating the interactions between MYOD/Myogenin and ARP5. Pulldowns with purified recombinant proteins isolated from bacteria or in vitro-translated proteins that do not have other mammalian factors would better support that the interaction is direct.

We performed co-immunoprecipitation assay using purified Arp5 and MyoD and confirmed the direct interaction between them (Figure 5A).

2. Figure 1C and D should show the expression level of ectopic ARP5.

We showed the expression level data of ectopic Arp5 in the infected skeletal muscle tissues (Figure 1C).

3. Figure 4F: How was endogenous Myogenin distinguished from transfected Myogenin expression? The amount of ARP5 over-expression should be shown for this experiment.

To detect the expression of the endogenous Myog gene as distinct from the exogenous one, a reverse primer for real time RT-PCR was designed for the 3’ UTR of Myog gene that is not included in the exogenous Myog mRNA (Table 1; p20, line 451- p21, line 455). The expression data of Arp5 was added in Figure 4H (which was Figure 4F in the previous version).

4. Although there is a statement in Methods regarding the number of independent experiments for statistical analysis, it is not clear how many experiments, representative figures like westerns, and other non-quantitative data represent. This should be clearly stated.

For non-quantitative data such as western blotting, all experiments were independently replicated three times, and representative images are shown in the figure. This statement was added in Materials and methods (p25, line 563-566).

5. What is the negative control in Figure 7B? In general, the manuscript would benefit from more experimental details in the Methods section and in figure legends.

Nonimmune IgG was used in the control experiments. This was described in figure legend and Materials and methods (p23, line 516).

6. The sentence on page 10, line 211 which states that ARP5 affects the expression of MYOD target genes in a BRG1-dependent manner is an overstatement. The data from the microarray is correlative and the ChIP data in Figure 7 indicate an association with effects on BRG1 recruitment. It is still possible that ARP5 can have repressive effects on muscle specific gene expression by additional mechanisms that do not involve BRG1.

In response to the reviewer’s comment, the wording of the sentence has been changed as follows. “These data raise the probability that Arp5—and not Ies6 or Ino80—affects the expression of MyoD target genes in a Brg1-dependent manner.” (p11, line 222-224)

7. A schematic model of the regulatory events would help summarize the findings.

Schematic representation of the role of Arp5 in myogenic gene expression was shown in Figure 7 supplement 2.

Reviewer #3:[…]1. A recent paper in EMBO J looked at the role of Ino80 as an interactor of Linc-Myh in the control of myoblast proliferation. In that study, Ino80 was required for myoblast proliferation. Here the authors have excluded the possibility that ARP5 is acting through Ino80 using qPCR of myogenic target genes in RMS cells after siRNA knock-down. The effect of knocking down ARP5 and Ino80 should be examined in proliferating C2C12 cells to see if they are acting through different pathways. Would overexpression of ARP5 continue to block differentiation if Ino80 was knocked down?

As mentioned by the reviewer, Schutt et al. (2020) reported that INO80 and Linc-Myh cooperatively regulate muscle stem cell proliferation through YY1 and/or p53 signaling pathway. The expression of Linc-Myh is muscle specific, whereas INO80 is ubiquitously expressed and necessary for the cell cycle progression of both muscle and non-muscle cells. In RD cells, we demonstrated that knockdown of Ino80 and Ies6 significantly altered the expression levels of genes related to p53 signaling pathway and cell cycle, whereas knockdown of Arp5 did not (Figure 2 supplement 1). Particularly, the expression of p21 (Cdkn1a), which is a main target of p53 in cell cycle regulation, was markedly increased in Ino80-si cells compared to Arp5-si cells (Figure 2 Table supplement 1). These are in contrast to the changes in the myogenic gene expression. Further, Arp5-KO cells normally proliferate under the culture conditions (Figure 3C). All these results indicate that Arp5 regulates the myogenic gene expression through a pathway different from that of the Ino80-mediated cell cycle regulation. We added these considerations in this manuscript (p14, line 310 – p15, line 321). We also performed a promoter-luciferase assay under the Ino80-knockdown conditions and confirmed that Arp5 inhibits the activation of myogenin promoter by MyoD independently of Ino80 (Figure 4F).

2. The authors have made a link between ARP5 expression and rhabdomyosarcoma tumor growth based on the over expression of ARP5 in the RMS cell line. While these results are convincing, it would be important to show the general interest of these findings in RMS. Indeed, many different factors have been shown to prevent tumour growth when knocked-down in RMS cells. The authors should examine RNA-Seq data of human rhabdomyosarcomas from the GEO data (or others) to see whether ARP5 is regularly over expressed in rhabdomyosarcomas compared to myoblasts.

Figure 2A shows RNA-seq data of human rhabdomyosarcomas obtained from GEO database, which revealed that Arp5 expression was more abundant in rhabdomyosarcomas than in normal skeletal muscle and tumor adjacent skeletal muscle.

3. The authors should examine the knockdown of ARP5 in proliferating C2C12 cells to determine the importance of ARP5 to blocking the myogenic differentiation activity of MyoD in myoblasts.

We performed Arp5 knockdown experiments in C2C12 cells and showed that Arp5 knockdown increased the expression level of myogenic genes even under the growth conditions, although myotube formation was not observed (Fig, 1H; p5, line 81-84).

4. It would be nice to see ChIP-seq results for MyoD with and without ARP5 knockdown to assess whether MyoD binding is specifically lost at Pbx-Meis binding sites.

We performed ChIP-seq analysis using anti-MyoD antibody in WT and Arp5-KO RD cells and demonstrated that Pbx-Meis-binding motifs were significantly enriched in the peak sequences from Arp5-KO cells (Figures 7D and 7E).

5. The effects of ARP5 overexpression and knock-down should be confirmed in primary myoblasts.

We performed gain and loss of function experiments of Arp5 in primary mouse myoblasts, and confirmed that Arp5 plays repressive roles in the expression of myogenic genes also in these cells (Fig, 1H; p5, line 81-84).

6. In figure 5e, the panel showing the ARP5 western lots in the pull down are missing. These should be added to the image.

We added the western blot images of ARP5 in Figure 5F (which was Figure 5E in the previous version). Since Arp5 does not bind to DNA, no band of Arp5 protein was detected.

[Editors’ note: what follows is the authors’ response to the second round of review.]

The manuscript has been improved but there are some remaining issues that need to be addressed, as outlined below:Reviewer #1:In this new version of the study, the authors have addressed many of the issues raised in the original version. The data showing that ARP5 can compete MYOD-MEIS/PBX binding and weakened MYOD interaction with DNA and chromatin has now been addressed in a much more convincing way. Specifically, the authors have performed MYOD ChIP-seq in the ARP5 KO cells. They claim that binding of MYOD is increased in absence of ARP5 and new sites, or sites with strongly increased occupancy are observed. However, the presentation of this data is sparse. The authors should show the normalized meta-profiles of MYOD binding to commonly occupied sites in each condition. They should show read density heat maps for total non-redundant sites in each condition to illustrate specific MYOD binding to sites only in the absence of ARP5.

We show meta-gene profiles and density heatmap of WT and Arp5-KO reads in Figure 7D and 7E. These data indicate that MyoD-binding sites are enriched near the transcription start sites in both WT and Arp5-KO RD cells. There is no noticeable difference between these meta-profiles, while Arp5-KO reads are more concentrated around TSS of some genes in the heatmap. (p12 line248 – line251, p24 line538 – line 540, page37 line 838 – line840, Figure 7D, E).

The total number of sites detected in each condition should be shown and their genomic distributions illustrated.

The total numbers of peaks were described in the revised manuscript, and their genomic distribution was illustrated in Figure 7F. (page12 line251 – line256, page37 line840 – line 841, Figure 7F).

DNA motif analyses should be shown on sites where no change is seen in absence of ARP5 compared to sites that show enhanced MYOD occupancy.

Motif enrichment analysis of the unchanged KO-peaks was performed. The Pbx–Meis-binding motifs were also identified in the KO-unchanged peaks, but the statistical significance of their enrichment were relatively lower than those of the KO-predominant peaks. These data were added to Figure 7H. (page13 line 270 – line273, page37 line844 – line845, Figure 7H).

Figure 7F is very difficult to interpret. The authors should find a much better and more quantitative method for integrating these different data sets.A more quantitative analysis of the data will strengthen the authors conclusions.

We show the percent overlap between KO-predominant peaks and MyoD, *MEF2*, Pbx, and Brg1 ChIP-seq peaks from the published database in Figure 7I (revised version of Figure 7F). Statistical analyses were performed by Fisher’s exact test using mononucleotide-shuffled sequences of the KO-predominant peaks as a control. This figure clearly shows that the KO-predominant peaks highly significantly overlap with MyoD1, Pbx, and Brg1-binding sites compared to the control sequences. (page13 line277 – line281, page37 line845 – line850, Figure 7I).

Reviewer #3:The authors have done extensive new experimentation to address my previous concerns. These experiments have solidified the authors conclusions that Arp5 competes with Pbx/Meis for binding to the CR domain of MyoD to regulate function.Page 6, line 104-105 the authors state "These results indicate that Arp5 overexpression contributes to MRF dysfunction in RMS."This is an over interpretation of the results to this point. Please change to "These results suggest that Arp5 overexpression may contribute to MRF dysfunction in RMS."

We have revised the text as suggested by the reviewer. (page6 line104 – line 105).